# Spatial frequency adaptation modulates population receptive field sizes

Ecem Altan[1]*, Catherine A Morgan[2,3], Steven C Dakin[1,4], D Samuel Schwarzkopf[1,5]*

[1]School of Optometry and Vision Science, The University of Auckland, Auckland, New Zealand; [2]School of Psychology and Centre for Brain Research, University of Auckland, Auckland, New Zealand; [3]Centre for Advanced MRI, University of Auckland, Auckland, New Zealand; [4]UCL Institute of Ophthalmology, University College London, London, United Kingdom; [5]Experimental Psychology, University College London, London, United Kingdom

## eLife Assessment

This well-designed study combining psychophysical and fMRI data presents a **valuable** finding regarding how adaptation alters spatial frequency processing in the cortex. The evidence supporting the claims of the authors is **solid**, although inclusion of more participants and better quality of the fMRI data would have strengthened the study. The study will be of interest to cognitive and perceptual neuroscientists working on human and non-human primates.

*For correspondence:
altan.ecem@hotmail.com (EA);
s.schwarzkopf@auckland.ac.nz
(DSS)

**Competing interest:** The authors declare that no competing interests exist.

## Abstract

The spatial tuning of neuronal populations in the early visual cortical regions is related to the spatial frequency (SF) selectivity of neurons. However, there has been no direct investigation into how this relationship is reflected in population receptive field (pRF) sizes despite the common application of pRF mapping in visual neuroscience. We hypothesised that adaptation to high/low SF would decrease the sensitivity of neurons with respectively small/large receptive field sizes, resulting in a change in pRF sizes as measured by functional magnetic resonance imaging (fMRI). To test this hypothesis, we first quantified the SF aftereffect using a psychophysical paradigm where human observers made SF judgments following adaptation to high/low SF noise patterns. We then incorporated the same adaptation technique into a standard pRF mapping procedure to investigate the spatial tuning of the early visual cortex following SF adaptation. Results showed that adaptation to a low/high SF resulted in smaller/larger pRFs, respectively, as hypothesised. Our results provide the most direct evidence to date that the spatial tuning of the visual cortex, as measured by pRF mapping, is related to the SF selectivity of visual neural populations. This has implications for various domains of visual processing, including size perception and visual acuity.

## Introduction

Neurons in the visual cortex respond to stimulation within a limited range of the visual field, namely the receptive field. The receptive fields are organized in such a way that adjacent neurons are stimulated by adjacent areas in the visual field (*Wandell et al., 2005*; *Wandell and Winawer, 2011*). A consequence of such retinotopic organisation is that receptive field characteristics of the human visual cortex can be studied with a model-based pRF mapping method (*Dumoulin and Wandell, 2008*) using fMRI. The method involves systematically stimulating different regions of the visual field and modelling the corresponding fMRI responses to estimate quantitative pRF properties, such as size and position.

Both pRF mapping (*Dumoulin and Wandell, 2008*; *Amano et al., 2009*; *Dumoulin and Harvey, 2012*) and electrophysiology (*Van Essen et al., 1984*; *Klink et al., 2021*) studies have established that the receptive field sizes of neuronal populations depend on (a) the eccentricity of the receptive field location and (b) the visual area they are drawn from, within the visual hierarchy. Neurons corresponding to the central visual field exhibit smaller receptive fields (finer spatial tuning), whereas those responding to the more peripheral visual field have larger receptive fields (coarser spatial tuning). The dependence of receptive field size on eccentricity has been linked to human visual performance, specifically visual acuity (the smallest size of object we can reliably recognise) and discrimination of relative size. A progressive increase in eccentricity is associated with a proportional decline in resolution visual acuity (*Anstis, 1974*). Additionally, perceived size biases in eccentric areas of the visual field are linked to the spatial tuning pattern across eccentricity (*Moutsiana et al., 2016*; *Urale and Schwarzkopf, 2023*).

Neurons in early visual areas are selectively sensitive to a limited range of spatial frequencies (SFs) within their receptive fields. SF tuning in the visual cortex has been reported in cats (*Everson et al., 1998*; *Hübener et al., 1997*), macaque monkeys (*Xu et al., 2007*; *De Valois et al., 1982*), and humans using fMRI (*Aghajari et al., 2020*; *Henriksson et al., 2008*; *Broderick et al., 2022*). This selectivity has also been probed psychophysically using an adaptation paradigm developed by *Blakemore and Campbell, 1969* (also see *Campbell and Robson, 1968*; *Blakemore and Sutton, 1969*; *Blakemore et al., 1970*). Prolonged exposure to a grating of a particular SF (1) reduces perceptual sensitivity to the adapted SF, which manifests as an increase in the detection threshold for a grating at the adapted SF; and (2) alters the apparent SF of subsequently presented gratings such that their apparent SF is 'repelled' from the adapted SF (i.e. lower SF gratings appear even lower, higher SF gratings appear even higher; see *Webster, 2015* for a review on visual adaptation). This adaptation effect suggests the existence of SF tuning channel units, each maximally sensitive to a given relatively narrow range of SFs. A decline in one channel's sensitivity results in the adjacent channels taking the lead in the overall response, thereby shifting the perceived SF (*Braddick et al., 1978*; *Mollon, 1974*; *Frisby and Mayhew, 1980*). A similar channel mechanism has been proposed to account for eccentric size aftereffects (*Altan and Boyaci, 2020*) and numerosity adaptation (*Aulet and Lourenco, 2023*), although others have claimed SF channels themselves play a direct role in numerosity judgments (*Dakin et al., 2011*; *Paul et al., 2022*). Moreover, studies have suggested that SF-tuned mechanisms play a crucial role in size perception (*Carrasco et al., 1986*).

The neuronal SF preference across eccentricity follows a similar pattern to the variations in the receptive field size with respect to eccentricity. Neurons responding to peripheral stimulation are typically tuned to lower SFs than those representing central vision, and vice versa (*Aghajari et al., 2020*). In addition, SF-tuned neurons in the primary visual cortex are roughly scaled versions of one another in terms of their preferred SF, suggesting scale-invariance in neuronal processing of SF (*Teichert et al., 2007*; also see *Chen et al., 2020*). Based on these observations, one might infer a link between the SF selectivity of neuronal populations and their receptive field sizes. Such a connection is supported by single-unit recordings from the striate cortex of macaques (*De Valois et al., 1982*; *Foster et al., 1985*) and cats (*Movshon et al., 1978*; *Linsenmeier et al., 1982*). Furthermore, extending these findings (*Keliris et al., 2019*) introduced a novel method for estimating neuronal receptive field sizes in humans through fMRI, utilizing the SF selectivity of neurons. The results they provided were in close alignment with those obtained from electrophysiological measurements, yet they diverged significantly from pRF estimates.

While pRF estimation does not yield a direct measurement of neuronal receptive fields, it remains a widely employed technique in human research, spanning a diverse array of studies, including, but not limited to mapping the visual field (*Dumoulin and Wandell, 2008*; *Amano et al., 2009*), identifying neural correlates of visual performance and functioning (*Silva et al., 2021*; *Harvey et al., 2015*; *He et al., 2019*; *Shen et al., 2020*; *Vo et al., 2017*; *Dekker et al., 2019*), and examining how visual processing in the patient population diverges from the healthy population (*Papanikolaou et al., 2014*; *Alvarez et al., 2020*; *Schwarzkopf et al., 2014*). This prevalence underscores the importance of understanding how pRF measurements are connected to certain neuronal properties, such as SF selectivity, in addition to studying neuronal receptive fields directly. To our knowledge, no existing study has elucidated how the interplay between neuronal RF size and SF selectivity is reflected in pRF estimates.

In this study, we employed an SF adaptation paradigm within the standard pRF mapping design to explore this relationship in human pRFs. We hypothesized that SF adaptation would influence measured pRF sizes. As illustrated in *Figure 1a*, we anticipated that neurons would show decreased responsiveness after adaptation to their preferred SFs, and as a result, the sizes of the pRFs would be in accordance with those of non-adapted neurons. More specifically, we expected to observe larger pRF sizes following adaptation to high SF and smaller pRF sizes after adaptation to low SF (A simulation showing the hypothesized pRF size effects with limited contribution from either large or small RFs is included in the shared data repository). We first psychophysically tested the perceptual aftereffect of adapting to relatively high or low SF. Then, we examined the spatial tuning of the early visual cortex under the influence of these adapters.

## Psychophysics: spatial frequency aftereffect

The objective of the psychophysics experiment was to test whether the SF aftereffect, as documented in the literature (*Blakemore and Sutton, 1969*; *Blakemore et al., 1970*), could still be replicated using a brief top-up adaptation with bandpass-filtered isotropic noise images. While visual aftereffects are usually induced by long periods of initial and top-up adaptations to maximize the strength of the perceptual effect (*Greenlee et al., 1991*), our overarching aim was to integrate adaptation within the pRF mapping method. This necessitated the implementation of brief top-up adaptations as the pRF mapping design requires that each volume in the fMRI experiment include a top-up adaptation along with the mapping stimuli. Consequently, to ensure that the SF adapter stimuli designed for the pRF mapping context would induce a robust perceptual aftereffect, we investigated the effect of brief adaptation to noise images with either relatively high or low SF structures on the subsequent perception of mid-SF.

### Methods

#### Participants

Ten participants (three males and seven females; age range: 25–42; mean [$M$] = 36) with normal or corrected-to-normal vision participated in the experiment. The protocols and procedures were approved by the University of Auckland Human Participants Ethics Committee (reference 024231). All participants gave their informed consent prior to the experiment.

#### Stimuli and apparatus

Stimuli were presented via MATLAB (Mathworks) and the Psychophysics toolbox (*Brainard, 1997*). Participants used a chin rest to stabilize their heads 298 cm in front of a 4 K 65-inch monitor (LG webOS TV OLED 65E6T, 60 Hz refresh rate, 4096 × 2160 resolution). The gamma of the display was linearized in software using luminance estimates made with a photometer (LS100, Konica Minolta, Japan). The background luminance was $147 cd/m^2$.

Noise images were convolved with a log Gabor filter to pass a narrow range of SFs and fluctuations in local contrast of the pattern were flattened using the method described in *Dakin and Turnbull, 2016*. The stimuli consisted of two-dimensional normally distributed noise patterns that were isotropic and band-pass filtered to include only a narrow SF range. The contrast of the images was 100%.

*Figure 1b* shows an example illustration of the high-SF adapter stimulus, the mid-SF reference stimulus, and a test stimulus. There were two adapter conditions. One used low SF noise, which had a peak SF of 0.5 cpd, and the other used high SF noise with a peak SF of 3.5 cpd. The test phase included two noise stimuli. The one in the same visual hemifield as the adapter was called reference, and its peak SF was always ~ 1.3 cpd, the geometric mean of the SFs of the two adapters. The one in the unadapted visual hemifield, however, varied in SF throughout the experiment. There were 30 different SFs for the test stimulus, ranging from 0.2 to 3.8 cpd. During stimulus presentation, we alternated different noise images of the same SF at 20 Hz to prevent afterimages. For this purpose, we generated 40 noise images for each of the two adapter stimuli (high and low SF), 20 images for the reference stimulus, and 6 images for each of the 30 available test SFs. For all types of stimuli, the bandwidth of the peak spatial frequencies was 0.5 octaves.

Stimuli were presented within a circular aperture with a radius of 7.7° of visual angle. In addition, a gray mask occluded part of the stimulus. The mask consisted of a circle (radius: ~ 1° of visual angle)

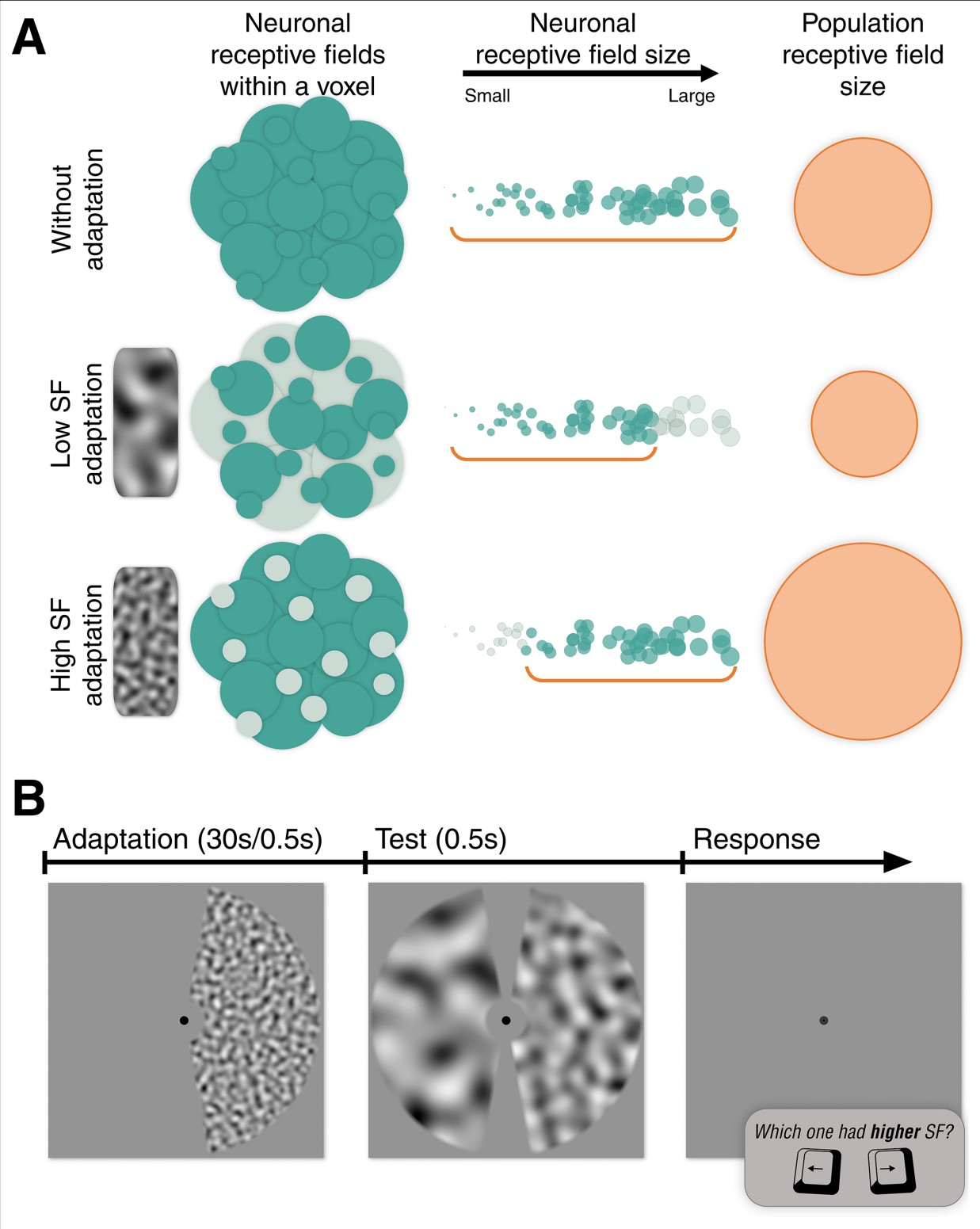

**Figure 1.** Hypothesized effect of SF adaptation on pRF size and stimulus sequence of the behavioural experiment. (**A**) Illustration of the hypothesized effect of SF adaptation on pRF size. Low and high SF adaptation decreases the responsiveness (indicated by paler green circles) of neurons having respectively large and small receptive fields, and the consequent pRF size estimations (orange circles) reflect the aggregate of the neurons that are not affected by the adapter (green circles). (**B**) Stimulus sequence of a single trial in the behavioural experiment. The adaptation phase included an adapter stimulus with high or low SF, either in the right or left visual hemifield (only present in adaptation conditions). The test phase consisted of a mid-SF reference stimulus on the adapted side and a test stimulus on the non-adapted side, which varied in SF, subject to two interleaved adaptive staircases.

*Figure 1 continued on next page*

*Figure 1 continued*

An empty display was shown until the participant responded. Participants were required to maintain their fixation on the dot at the center at all times and judge the relative SF of the two stimuli shown in the test phase. SF: spatial frequency.

at the center of the screen and two 20° wide wedges covering the top and bottom vertical meridian. A 0.4° wide alpha gradient mask was applied along the edges of the stimuli to eliminate the high SF components of the sharp edges. The stimuli for the psychophysics experiment were designed to match those of the fMRI arm of the study, where we aimed to avoid stimulating the contralateral hemifields.

## Procedure

Participants were first familiarized with the concept of spatial frequencies. They were shown a noise pattern similar to those used in the actual experiment and were allowed to interact with its SF using the H and L keys on the keyboard to make its SF gradually higher or lower, respectively. They were free to spend as much time as needed until they felt comfortable with the spatial frequencies and were ready for the experiment. Participants pressed the space bar to continue with the practice trials. There were 25 practice trials in which the participants were shown the test stimuli as they appeared in the actual experimental trials (see test phase in *Figure 1b*) and asked to indicate the one with higher SF. They were free to end the practice trials after they were comfortable with the task.

A single trial of the experiment consisted of the following phases: Adaptation (only in adaptation conditions), test, and response. Participants were required to fixate on a black dot at the center of the screen throughout the experiment. In the adaptation phase, participants were shown a ring segment of alternating noise images having a certain peak SF (either 0.5 or 3.5 c/deg). The adapter stimulus was either in the left or right visual hemifield. The adaptation lasted for 30 s in the first trials and 0.5 s in the rest of the trials, as a top-up adaptation.

The SF of the test stimuli was adaptively determined throughout the experiment. We used two interleaved one-up one-down adaptive staircases, starting from high and low SFs (3.8 and 0.2 cpd). The step size of the staircase was 0.5 cpd at the beginning, and it was halved after each response reversal until it reached 0.13 cpd. There were 30 trials in a staircase. The test phase was followed by a blank gray display, which signalled the participant to press one of the two arrow keys to indicate the location of the noise segment having a higher SF. The experiment proceeded to the next trial if one of the allowed keys was pressed within 2 s; otherwise, the trial was repeated.

An experimental session consisted of six blocks for three adaptation conditions (high SF adaptation, low SF adaptation, and no adaptation) and two visual hemifields (left and right). Sessions always started with two no-adaptation blocks in which the adaptation phase was absent, in order to avoid the influence of a possible long-lasting adaptation effect. The order for the rest of the blocks was randomly presented. Participants took short breaks between blocks.

## Results

All participants completed multiple sessions, ranging between 2 and 4 ($M = 2.8$) depending on their availability. Data gathered from the repeated sessions and also from the two visual fields were pooled and fitted with a log-normal function, using Psignifit 4 toolbox (*Schütt et al., 2016*) in Matlab, which resulted in three psychometric functions for each participant (from the three adaptation conditions). *Figure 2a* shows the psychometric functions of a representative participant. We determined the point of subjective equality (PSE) as the SF corresponding to the half proportion of the fitted function. Here, the PSE indicates the SF of the test noise that appeared equal to that of the reference noise. PSE values are the natural logarithm of SF in cycles per degree. Statistical analysis was performed with JASP (*JASP Team, 2023*).

On average, the perceived SF of the reference stimulus (compared to the veridical mid-SF) was 7% higher, 13% lower, and 0.03% higher, respectively, under the low SF adaptation, high SF adaptation, and no-adaptation conditions (see *Figure 2b*). To test whether these differences were significant, three one-sample $t$-tests were performed with the log-transformed SF ratios (natural logarithm of perceived/actual SF). p-Values were Bonferroni corrected for multiple comparisons. This revealed a significant difference from zero in both the low-SF ($t(9) = 3.8$, $p = 0.01$) and the high-SF adapted condition ($t(9) = -6.9$, $p < 0.001$). The no-adaptation condition did not significantly differ from zero

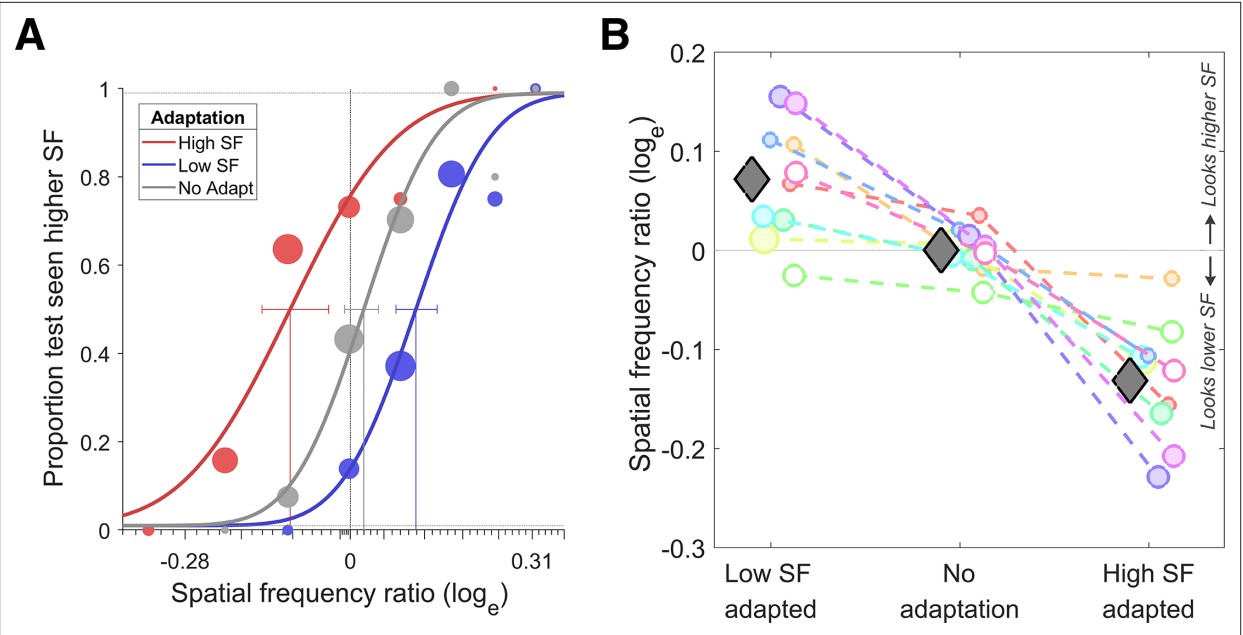

**Figure 2.** Results of behavioral experiment: spatial frequency aftereffect. (**A**) Psychometric functions for a representative participant. The *x*-axis represents the log-transformed SF ratio (test/reference SF), where negative values indicate a decrease and positive values indicate an increase in perceived SF. The *y*-axis represents the proportion of trials participants perceived the test stimuli as having a higher SF. (**B**) Log transformed SF ratios (perceived/actual) of all participants across three adaptation conditions. Coloured circles represent the average of left and right adaptation conditions for each participant, and the gray diamonds show the group average. Positive values on the *y*-axis denote perceptual overestimation of SF, while negatives denote the underestimation of SF. Unshaded circles indicate the participants who were not included in the fMRI arm of the study. Circle sizes are proportional to the number of sessions completed. SF: spatial frequency.

($t(9) = 0.05$, $p = 0.9$). Assuming that the PSEs from the no-adaptation condition reveal any potential response bias of the observers and that the adaptation conditions suffer from the same response biases, we corrected for the response bias (separately for left and right adaptation conditions) by subtracting the PSEs of the no-adaptation condition from the corresponding PSEs of the adaptation conditions. One sample *t*-tests for bias-corrected adaptation conditions again revealed a significant difference from zero (low-SF adapted: $t(9) = 4.3$, $p < 0.01$; high-SF adapted: $t(9) = -5.7$, $p < 0.001$; Bonferroni corrected). Additionally, we found that the illusion magnitudes were not significantly different between the two visual hemifields (see appendix A.6 'Does the perceptual effect differ between the two visual hemifields?').

In conclusion, results showed strong adaptation effects on the perceived SF in both high and low SF adaptation conditions. Our behavioural results are consistent with previous studies on SF adaptation (*Blakemore and Sutton, 1969*; *Blakemore et al., 1970*), as well as a variety of other studies on visual adaptation to size (*Altan and Boyaci, 2020*; *Pooresmaeili et al., 2013*), shape (*Suzuki and Cavanagh, 1998*; *Storrs and Arnold, 2017*), texture density (*Sun et al., 2017*), and motion (*Mather et al., 2008*), in terms of the bidirectional repulsive shift in perception of the adapted stimulus feature. Confirming that both of the adapter stimuli we used effectively altered the perceived SF, we used the same SF adapters in the fMRI experiment.

## fMRI: pRF size change following spatial frequency adaptation

The purpose of the fMRI experiment was to investigate the effect of SF adaptation on the receptive field size of neuronal populations in early visual areas. Assuming larger RFs convey selectivity for low SF and smaller RFs convey selectivity for high SF, we expected to find relatively smaller pRFs after low SF adaptation and larger pRFs after high SF adaptation (see *Figure 1a*). This is because the adaptation would decrease the responsiveness of a certain group of neurons that are selectively sensitive to the adapter SF, and this would lead to a change in the overall pRF size, favoring the relatively non-adapted neurons. A simulation of the adaptation effect on pRF size can be found in the shared data repository.

## Methods

### Participants

We only scanned the participants who showed a strong SF aftereffect in the behavioural experiment because we were interested in studying the properties of neural populations underlying the perceptual adaptation effect. The participants we recruited showed a clear separation between the psychometric functions for the two adaptation conditions (blue and red curves in *Figure 2a*), where the 95% confidence intervals of the PSE measurements do not overlap. In this regard, only one participant, shown with the unshaded green circles in *Figure 2b*, was excluded due to not demonstrating a strong SF aftereffect. Additionally, another participant, represented with the unshaded pink circle in *Figure 2b*, was excluded due to an MRI-incompatible metal implant. Therefore, 8 of the 10 volunteers from the psychophysics experiment participated in the fMRI experiment. All participants gave their informed consent prior to the experiment.

### Procedure and stimuli

Participants lay in the scanner bore with their nasion positioned at isocentre, and observed (via a mirror on the head coil) the stimuli presented on an MRI-compatible LCD monitor (BOLD screen, Cambridge Research Systems, Rochester, UK) placed at the back of the scanner bore. The viewing distance was 111 cm. Stimuli were generated and presented via MATLAB (Mathworks) and Psychtoolbox (*Brainard, 1997*). Eye movements were recorded via an MRI-compatible eye tracker (Eyelink 1000+, SR Research, Ottawa, Canada).

The stimulus sequence used in scanning sessions is illustrated in *Figure 3*. Each mapping run started with a 25 s baseline in which only the fixation point was shown on a gray background. This was followed by the initial adaptation period of 30 s. Adapter stimuli were exactly the same as in the behavioral experiment, except that the diameter of the circular mask at the center of the screen was 0.7° of visual angle, and there were two adapter segments presented on the left and right visual hemifields at the same time. One of the adapters consisted of the low SF (0.5 cpd), and the other consisted of the high SF noise images (3.5 cpd), throughout the run. The location of the high and low SF adapters was alternated between left and right visual hemifield in each consecutive run. This way, half of the runs included a high-SF adapter on the left and a low-SF adapter on the right visual hemifield (High-Low or HL runs), and the other half included the opposite configuration (Low-High or LH runs). There were short breaks (at least 30 s) between runs to allow washout of adaptation and also for participants to rest their eyes.

After the initial adaptation, the pRF mapping sequence started. The video containing the stimuli is available in the shared data repository. We used bar stimuli containing high-contrast noise images with mid-SF (~ 1.3 cpd; the geometric mean of the adapter spatial frequencies). The noise images on the bars alternated at 20 Hz. The bars were presented for 0.5 s and preceded by a 0.5 s of top-up adaptation (this makes the effective initial adaptation period 30.5 s). The stimuli were shown within a circular aperture with a diameter of 15° that is equal to the height of the display. Bars were also occluded in the unadapted areas via a gray mask. The contours of the bars constituted by the mask or the aperture were smoothed. The width of the bars subtended 1.15° of visual angle, unless occluded by the mask. The length also varied due to the mask and the aperture.

Twenty-five bars that were half-overlapping in width stepped through the screen in 25 s. The visual hemifield was swept by these bars eight times in a run. The first sweep was from bottom to top, and the following sweeps were 45° clockwise rotated versions of the previous one. In addition, there were 25 s mean-luminance blocks in the middle and at the end of the eight directions, where instead of a bar, a blank screen followed the top-up adaptations. The runs ended after a second 25 s baseline with nothing but a fixation point shown on a gray background.

The fixation point was presented at the center of the screen at all times. To measure participants' performance in fixating at the center, we changed the fixation color and asked participants to press a key (on an MRI-compatible response box) when the color changed. The probability of the fixation dot changing color from blue to green was 0.01 for each 200ms epoch of the total scanning time. The duration of the color change was 200ms.

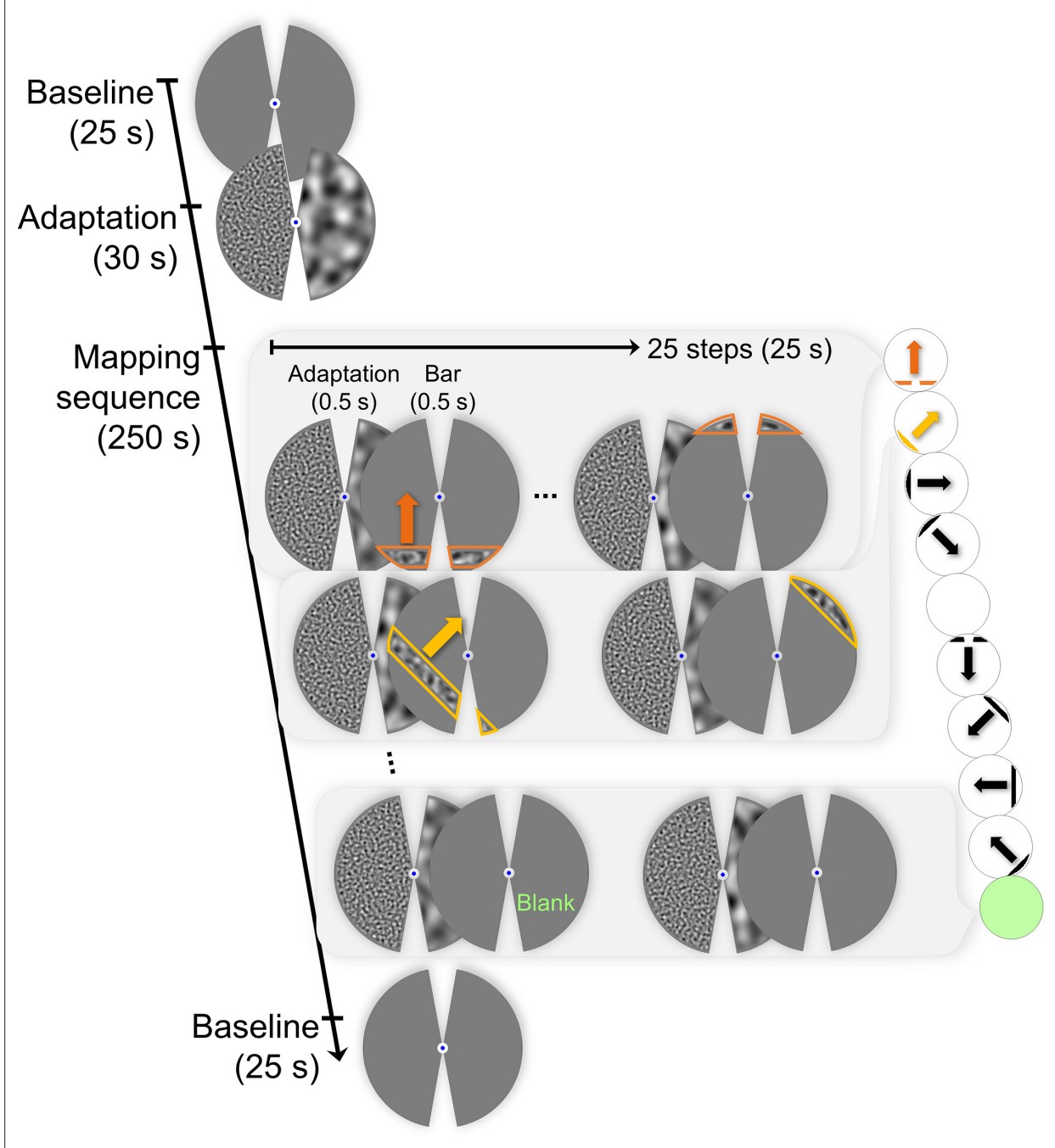

**Figure 3.** Stimulus sequence of a pRF mapping run. Scans started with a baseline period, showing an empty gray display. This was followed by the initial adaptation period. Adaptation consisted of two noise stimuli occupying both visual hemifields, having different SFs (in this case, high SF was on the left and low SF was on the right, i.e. HL configuration). During the mapping sequence, mid-SF bars swept the screen step by step, in eight directions. Two blank phases were presented in the middle of eight bar directions and also at the end of the mapping sequence. Each mapping component (bars and blanks) was preceded by a short top-up adaptation stimulus, consistently in all volumes throughout the mapping phase. Finally, the runs were concluded with another baseline. All noise stimuli were refreshed at 20 Hz, and a central fixation point persisted throughout. Participants pressed a key upon a color change in the fixation.

## Data acquisition

MR images were collected at the Centre for Advanced Magnetic Resonance Imaging (CAMRI) in the Faculty of Medical and Health Sciences, the University of Auckland. We used a 3 Tesla MAGNETOM Skyra MR scanner (Siemens Healthcare, Erlangen, Germany). The 32-channel head coil was used, but

with the front element removed to convey an unoccluded view of the stimulus, leaving 20 effective receive channels.

MRI sessions started with acquiring a T1-weighted image optimised for gray and white matter contrast with a TR of 2 s, a TE of 2.85ms, a TI of 880ms, a flip angle of 8°, and 1 mm isotropic voxel size. High-resolution structural images were collected to project the functional data onto. This was followed by functional runs for pRF mapping; each lasted 5 min 30 s. The total number of pRF runs varied between six to ten across participants but always maintained an even count. This ensured that each participant had an equal number of runs for both HL and LH configurations. The variation in the number of runs arose due to time constraints of the scanning sessions and occasional participant preferences to conclude the session earlier than scheduled.

The gradient-echo, T2*-weighted pRF mapping scans were acquired with 2.3 mm isotropic voxel size. Scans had a field of view of 221 mm, TR of 1 s, TE of 30ms, flip angle of 62°, a rBW of 1680 Hz/pixel, a multiband slice acceleration factor of 3, and an in-plane/parallel imaging acceleration factor of 2. We used 36 transverse slices angled to be approximately parallel to the calcarine sulcus that covered the whole occipital cortex.

## Preprocessing

We preprocessed the functional images using SPM12 (*Penny et al., 2007*). We corrected motion artifacts using the realign and unwarp module with the default parameters. Then we coregistered the realigned and unwarped functional images to the structural scan. To obtain a surface mesh of the pial and gray-white matter boundaries, the structural images were reconstructed with FreeSurfer (Version 7.1.1).

Preprocessed functional volumes were projected onto the surface mesh using FreeSurfer. We first detrended and normalized the time series in each vertex (each point in the surface mesh representing the projected functional data) by $z$-score. To quantify the reliability of the visual responsiveness of each vertex, we calculated the noise ceiling. For this, we first correlated the time series of odd and even runs of each condition, then we used the Spearman-Brown prophecy formula (*Brown, 1910*; *Spearman, 1910*) to calculate the reliability of the average of all runs of the same condition. The noise ceiling is equal to this reliability measure as it denotes a maximum achievable goodness of fit for each vertex. The details of these calculations can be found in *Morgan and Schwarzkopf, 2020* and *Urale et al., 2022*.

We then averaged the runs for each condition. This process yielded two hemisphere files for both the HL and the LH adapter conditions. We discarded the volumes that were not pertinent to the pRF estimation, such as the initial adaptation period and the two baselines at the beginning and at the end of the mapping sequence. Finally, the hemispheres from the same adapter condition were combined to produce a single file for the high/low SF adapted condition.

## pRF estimation and region of interest (ROI) identification

The data analysis was performed using the SamSrf toolbox, version 8.3 (*Schwarzkopf, 2021*) in MATLAB 2020b. We generated a binary representation of the bar stimulus at each TR and calculated each voxel's predicted response to the stimulus from the overlap with the voxel's modelled pRF profile. Here, the predicted response at each TR was calculated based on the bars (not adapters), because the visual region stimulated by the adapter was the same in all TRs, critically during the blank periods as well. Since the adapters provide a constant source of signal throughout the mapping sequence, their contribution to the signal cancels out over the time series and they cannot be modelled separately.

We modelled pRFs as two-dimensional Gaussian functions with these free parameters: Cartesian coordinates, $x_0$ and $y_0$, and the size of the pRF, which is defined as the standard deviation, $\sigma$ of the Gaussian function. Convolving the predicted pRF response with the canonical hemodynamic response function (HRF; determined based on previously acquired data *de Haas et al., 2014*), we predicted the Blood Oxygenation Level Dependent (BOLD) time series for each voxel. To find the optimal pRF parameters that minimize the error between the predicted and empirical BOLD time series, we followed a coarse-to-fine fit approach. First, we generated an extensive search grid with the numerous plausible combinations of the three pRF parameters. The predicted BOLD time series at each point in the search grid was correlated with the observed BOLD time series. The parameters that led to the best correlation were then used in the second stage of the fitting. We used the Nelder-Mead simplex

search algorithm (*Nelder and Mead, 1965*; *Lagarias et al., 1998*) to further refine the parameters. We estimated the optimum parameters based on the maximum correlation between the predicted and observed BOLD time series. At the final stage, we estimated the response amplitude and baseline parameters of this predicted time series via linear regression.

The time-intensive pRF analysis was restricted to relevant regions, namely the posterior portion of the cortex containing the occipital lobe. The optimized estimations for the pRF parameters were used to generate polar angle, eccentricity, and pRF size maps for both low and high SF adapter conditions (see *Figure 4*). We also calculated the goodness of fit ($R^2$) for each vertex by comparing the predicted and the observed time series and normalized it by dividing $R^2$ by the noise ceiling ($nR^2$).

For delineation purposes, we averaged the maps of the two adaptation conditions. We used the automatic delineation tool of the Samsrf toolbox to generate rough estimates of ROIs based on the default atlas (Infernoserpents) within the toolbox. We then manually fixed the automatically generated regions as described in the literature (*Wandell et al., 2007*; *Sereno et al., 1995*; *Wandell et al., 2005*;

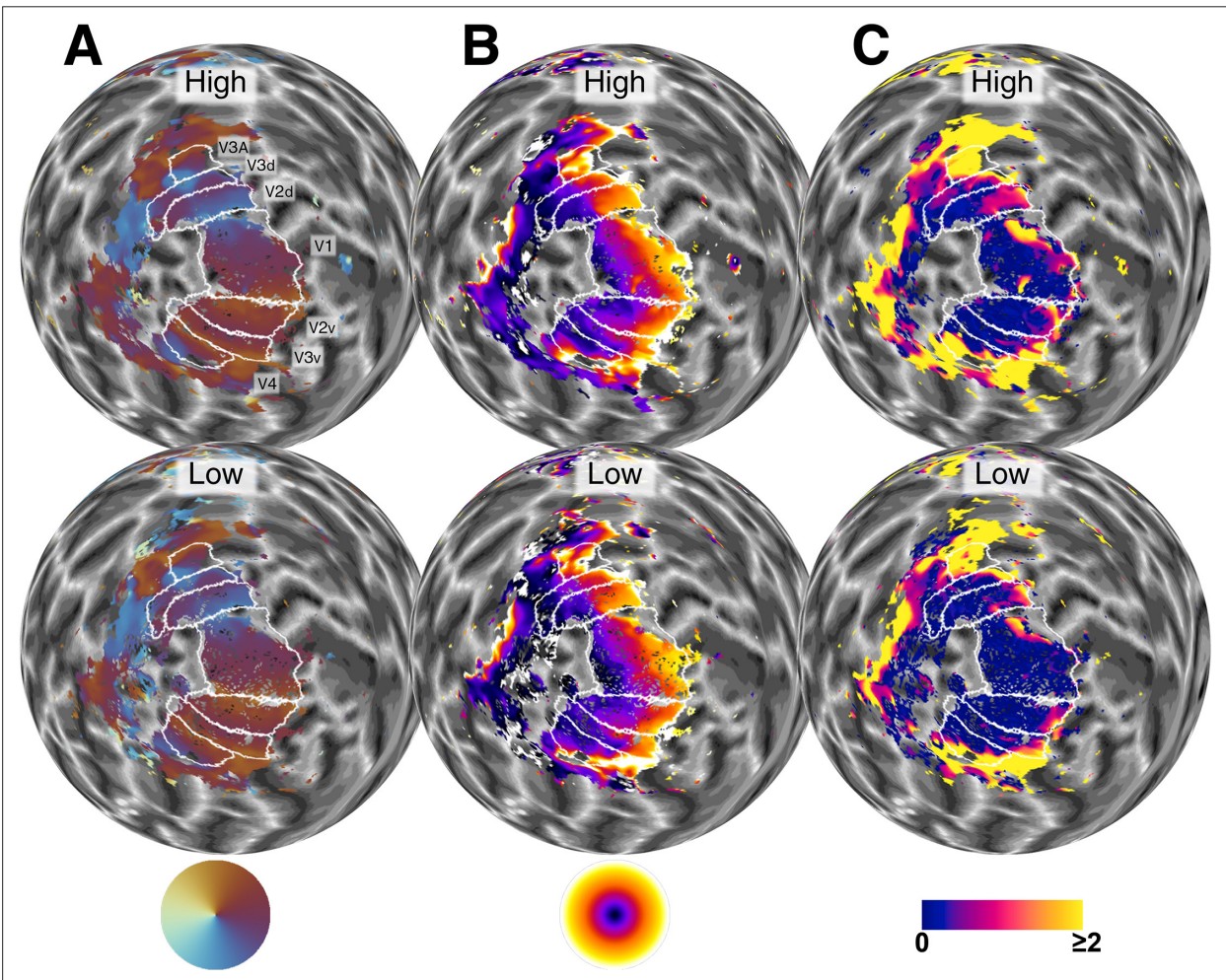

**Figure 4.** pRF Maps of a participant for high and low SF adapter conditions. Polar angle (**A**), eccentricity (**B**), and pRF size (**C**) maps for the left hemisphere of representative participant, on a spherical model of the gray-white matter boundary (see online data repository for other participants). The maps were plotted after the data were denoised and $nR^2$ thresholded, so that only the vertices used in the analyses would be shown (see text for thresholding details). The first row shows the maps from high-SF adapted condition, and the second row shows those from low-SF adapted condition; annotated as High and Low, respectively. Color charts of each map type are shown on the bottom row. Polar angle maps in column A show the angle from the fixation point. e.g. orange represents the vertical center of the upper visual hemifield. The eccentricity maps in column B show the distance from the fixation point, from 0° to 8° of visual angle, represented by the range of colors from black to white (inner to outer parts of the eccentricity color wheel). pRF size maps in column C show the standard deviation (sigma) values of the Gaussian pRF profile. Blue indicates the pRF sizes that are closer to zero, and yellow shows pRF sizes greater than or equal to 2° of visual angle. White lines on the maps represent the borders of visual ROIs, delineated using the average of the two conditions. The ROI labels are shown only in the first row of A for simplicity.

*Winawer and Witthoft, 2015*) and labeled five visual areas: V1, V2, V3, V4, and V3A. See the upper row of *Figure 4a* for the labeled regions for the left hemisphere of a representative participant.

Before statistical analyses, we denoised the pRF data to remove artifactual vertices where at least one of the following criteria was met: (1) sigma values were equal to or less than zero, (2) both $x_0$ and $y_0$ are zero, and (3) beta (amplitude) smaller than 0.01 or greater than 3. We also excluded vertices with weak goodness of fit, normalized $R^2$ smaller than 0.2. Because we expected that different sets of neurons would contribute to the individual voxel response in two conditions, due to adaptation, we did not match the removed vertices in conditions. However, to control for a possible systematic bias that could lead to comparing different regions of the brain, we also present results obtained by applying the same thresholding criteria and further limiting the analysis to the same vertices between conditions. *Appendix 1—figure 4a* presents the pRF size change using the data from the same vertices between conditions.

## Results

A visual inspection on the pRF size maps in *Figure 4c* clearly shows a difference between the two conditions, which is evident in all regions. The overall pRF sizes on the high SF adapted condition seem larger (less blue, more pink, and yellow) than those on the low SF adapted condition. pRF maps, including $R^2$ and $nR^2$, of all participants are available in the shared data repository. *Figure 5a* presents the median pRF sizes for all participants. On average, the median pRF sizes were larger in the high SF adaptation condition than in the low SF adaptation condition in all ROIs (gray diamonds).

To quantify the participant-wise pRF size difference in each ROI, we performed Mood's median test (independent samples) on the raw pRF size values between the two conditions. The reason for using this non-parametric median test is that the distribution of the pRF sizes is zero-bounded and highly skewed (see additional histogram plots in the online repository). Results of each participant-wise comparison are incorporated in *Figure 5a*, with filled circles indicating where there is a significant difference between the adapter conditions. In V1, six out of eight participants showed significantly larger pRF sizes in the high SF adapted condition as compared to those in the low SF adapted condition (V2: 8 of 8, V3: 5 of 8, V4: 6 of 8, and V3A: 4 of 8). The results were most consistent in V2, where all participants showed a significant effect in the expected direction. The effect is also consistent in V1 for the majority of participants. However, the results for the higher regions are more variable between participants.

As a group-level comparison, we performed a repeated measures ANOVA with the median pRF size as the dependent variable. There were two independent variables: (1) Region of interest (ROIs) with five levels (V1, V2, V3, V4, and V3A) and (2) Adapted SF with two levels (high and low). There were significant main effects of both ROI ($F(1.58, 11.06) = 21.63$, $p = 0.0003$; Greenhouse-Geisser corrected) and SF ($F(1, 7) = 14.48$, $p = 0.007$). There was no significant interaction between the two variables ($F(1.74, 12.17) = 1.97$, $p = 0.18$).

We also calculated the pRF size difference maps for all participants by subtracting the low-SF condition's pRF sizes at each vertex from the corresponding vertices in the high-SF condition. We binned the pRF difference data into 1° wide 100 eccentricity bins, using a sliding window approach. We used the average of the two conditions for eccentricity selection to mitigate binning bias, although we must note that using the average is not necessarily enough to control for this bias (*Stoll et al., 2022*). The median of each eccentricity bin was averaged across participants (see *Figure 5b*). The difference plots were consistently above zero along almost all eccentricity bins in all visual regions. This indicates that, on average, the median pRF size in the high SF adapted condition was larger than in the low SF adapted condition in all regions and eccentricities. (Also see *Appendix 1—figure 4b* which shows the pRF size of the two conditions separately, as a function of eccentricity).

As *Figure 5a* also shows, the pRF sizes vary considerably between visual fields (~ 0.1° in V1, ~ 2.5° in V3A), as an inherent feature of visual hierarchy. This makes the regions incomparable in terms of the pRF size change. Therefore, in addition to the absolute pRF size difference shown in *Figure 5*, we also calculated proportional pRF size difference maps for all participants (Details of this calculation are available in appendix A.2 'Proportional pRF size change map'). *Appendix 1—figure 2* shows the results for a representative participant whose pRF maps were shown in *Figure 4*.

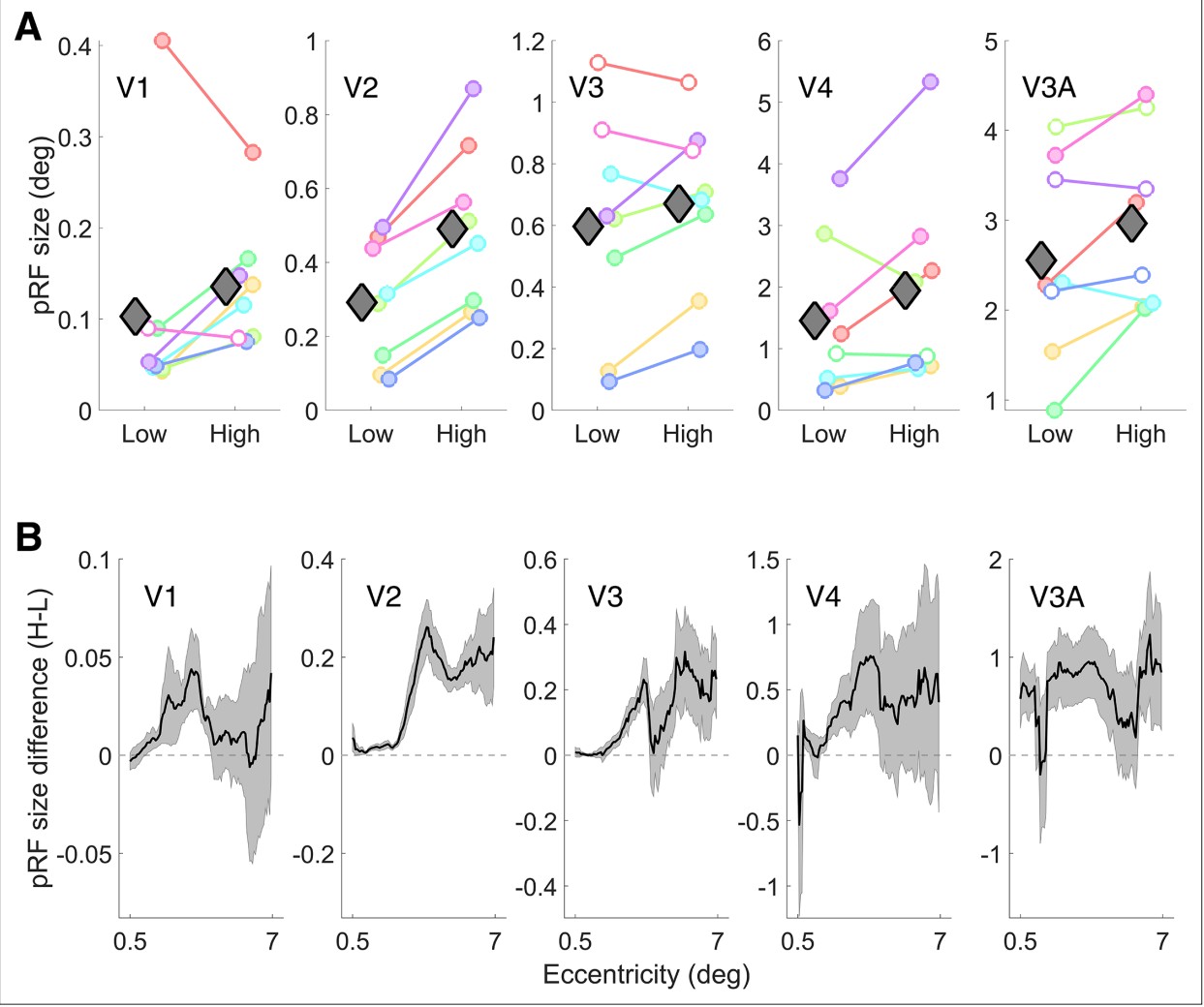

**Figure 5.** Median pRF size for both adapter conditions and pRF size difference as a function of eccentricity. (**A**) Median pRF sizes (*y*-axes) were plotted against low and high SF adaptation conditions (*x*-axes) for all ROIs (separate plots). Colored circles represent each participant. Filled circles indicate a significant participant-wise median comparison, with FDR corrected p < 0.05. Gray diamonds show the group means. (**B**) Median of eccentricity binned pRF size differences, averaged across participants. pRF size differences were calculated by subtracting the pRF sizes of the low SF adapted condition from those of high SF (H–L). Positive and negative values on the *y*- axis are related to respectively larger and smaller pRF sizes in the high SF adapted condition. The average of the two conditions was used for eccentricity binning. The shaded area represents the standard error of the mean between participants (N=8). The data illustrated in both panels were first denoised and thresholded as described in the text. Note different scales between regions.

Additionally, we tested possible signal-to-noise (SNR) ratio differences between the two conditions and found no consistent difference (see appendix A.3 'Signal to noise ratio' and *Appendix 1—figure 3a* for details).

The average time series of percent signal change highly matched between conditions. Noise ceiling values of each condition were also largely similar along eccentricity *Appendix 1—figure 3b*.

Lastly, to test whether the change in pRF sizes was accompanied by a change in pRF eccentricity, we plotted the eccentricity-binned eccentricity values from the two conditions (*Appendix 1—figure 4c*), and we did not observe any consistent change in pRF eccentricities between the two adaptation conditions (see appendix A.5 'Does SF adaptation alter other pRF estimates?' for more details). In addition, participant-wise comparisons of raw pRF estimates (polar angle, eccentricity, and pRF size) also show no difference between the conditions in terms of both eccentricity and polar angle. The figures are included in the shared data repository.

### Eye tracking and fixation task

The analysis for the performance of participants in responding to the fixation point color change showed that all participants successfully responded to most of the color events within 1 s. Percent correct responses among participants (averaged across runs) ranged between 74% and 99% ($M = 90\%$).

Eye tracking data showed no significant difference between the two adaptation conditions in terms of pupil size ($t(7) = 0.1$, $p = 0.9$) and distance of eye position from the fixation point ($t(7) = -0.14$, $p = 0.9$). Eye tracking data from two participants was excluded due to a suspected calibration error between runs. More details can be found in appendix A.7 'Eye tracking data'.

## Can perceived contrast explain the findings?

Stimulus contrast alters the receptive fields of visual neurons such that receptive fields (measured with mid-contrast grating stimuli) expand in response to low-contrast stimuli and contract in response to high-contrast stimuli (*Sceniak et al., 1999*; *Kapadia et al., 1999*). Although the physical contrasts of the high and low SF adapter stimuli were matched in our study, one may argue that the adapters can possibly change the perceived contrast of the mapping stimuli and that the perceptual contrast differences in the mapping bars could potentially affect pRF sizes. So lastly, to test whether this could influence our results, we conducted another psychophysical experiment. Here we tested if our SF adapters result in a change in perceived contrast, and critically, whether the perceived contrast differs between high and low SF adapter conditions. We used the same SF-filtered adapter stimuli from the fMRI experiment to test their possible effects on the perceived contrast of the subsequently presented mid-SF stimulus.

### Methods

#### Participants

Twelve participants (seven males, five females; age range: 23–44; $M = 30$) with normal or corrected-to-normal vision participated in the experiment. Five were recruited from the fMRI experiment. All participants gave their informed consent. The protocols and procedures were approved by the University of Auckland Human Participants Ethics Committee with reference number 024231.

#### Stimuli and procedure

The experimental procedure was the same as the initial psychophysics experiment. To familiarize participants with the stimuli and the contrast, a noise image was shown before the experiment. Participants were allowed to interact with the contrast of the image by pressing H and L keys. Then they proceeded to the practice trials when they felt comfortable. Practice trials consisted of 25 trials in which the participants were asked to judge the contrast of the two noise images and report the one with higher contrast. Practice trials were exactly the same as they would appear in the actual experiment's no-adaptation conditions.

Similar to the first psychophysical experiment, experimental trials consisted of an adapter (only in the adaptation conditions), a test phase, and a response period. The stimulus sequence of trials is illustrated in *Figure 6a*. The adapter and the reference noise images were exactly the same images that were used in the fMRI experiment, but presented similar to the first psychophysical experiment. The only difference in this second experiment was that we adaptively altered the contrast of the test stimulus in each trial, instead of the SF. We used two interleaved one-up one-down adaptive staircases. One of the staircases started from low contrast, which was 0.1; the other started from the maximum contrast, 1. Test stimuli were drawn from a set of 360 images: twelve noise images at each of thirty contrast levels (0.01–1.0 in steps of 0.031). The peak SF of these new images was 1.3 c/deg, the same SF as the reference images. Because this experiment aimed to test potential confound in the main experiment, the reference stimulus here always had the maximum contrast of 1, matching that of the mapping stimulus in the fMRI experiment. Note that we used high-contrast stimuli earlier to ensure a strong fMRI signal.

Unlike the SF experiment, here the reference stimulus had an equal stimulus level (i.e. contrast = 1) as the starting point of one of the staircases, so we were unable to test higher levels than the reference contrast. Fitting a psychometric function with this type of unilateral data is suboptimal, and in most cases, it results in inaccurate PSEs that are biased toward the untested values. Instead, we

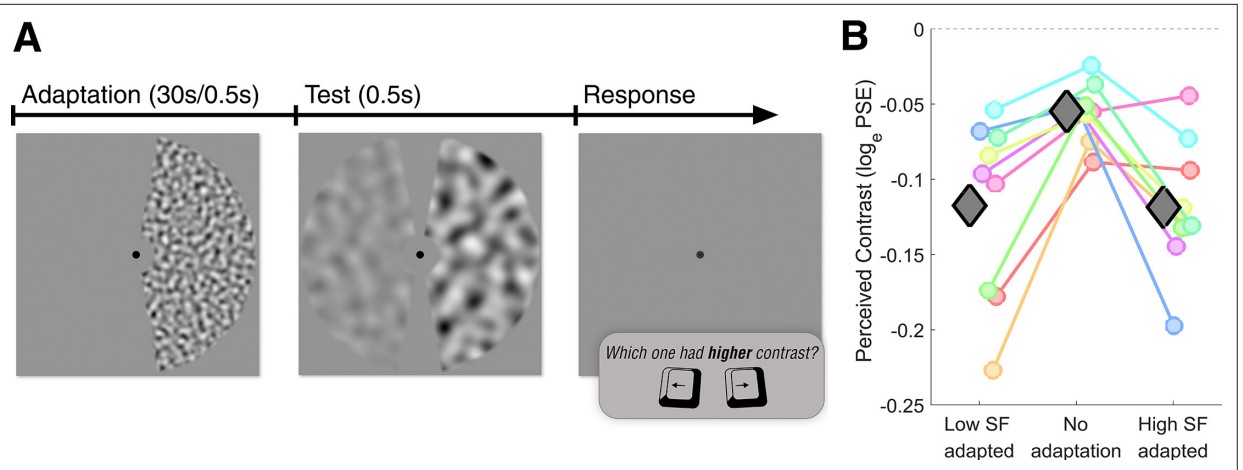

**Figure 6.** Control experiment: stimulus sequence and the effect of SF adapters on perceived contrast. (**A**) Stimulus sequence of a single trial in the control experiment. The adaptation phase is exactly the same as that in the main experiment. In the test phase, the adapted visual hemifield always included a mid-SF, maximum-contrast (100%) stimulus. The stimulus on the non-adapted visual hemifield had a mid-SF but varied in contrast, subject to two interleaved adaptive staircases. In the response phase, participants were required to indicate the stimulus with relatively high contrast. This sequence was repeated 60 times for each measurement. (**B**) Perceived contrast as a function of SF adaptation condition. Coloured circles represent the average log-transformed PSE values of each participant; gray diamonds indicate the participant average. SF: spatial frequency.

calculated the PSEs based on the response reversals. After removing the first three reversals, we averaged the contrast levels at each reversal, except for those ±2 median absolute deviations away from the median. Through visual inspection, we ensured the robustness of this method, which effectively removed transient deviations in the data and eliminated the staircases in which no response reversals were observed. Due to the insufficient number of response reversals (less than or equal to three) in both staircases of multiple measurements in any given condition, the entire data set from three participants was unusable.

## Results

Log-transformed PSEs of each repeated measurement of the same condition were averaged. *Figure 6b* shows the averaged PSEs of all participants for three adaptation conditions. We subtracted the response bias of each condition (PSEs of no adaptation condition) from the corresponding adaptation conditions, separately for left and right adapters. We performed statistical analyses on the bias-corrected log-transformed PSEs, using JASP (*JASP Team, 2023*).

Paired samples $t$-test showed no significant difference between the two adaptation conditions in terms of their effect on perceived contrast, as expected ($t(8) = 0.04$, $p = 0.97$). To test the statistical evidence for the null hypothesis, we also performed a Bayes factor hypothesis testing with a two-sided Bayesian paired-samples $t$-test. We used an uninformed prior distribution which is the default Cauchy distribution with $r = 1/\sqrt{2}$. The result showed moderate evidence for the null hypothesis which suggests no difference between the two adaptation conditions in terms of their effect on perceived contrast ($BF_{null} = 3.1$, error percentage = 0.004%). Results suggest that the change in receptive field size we observed after SF adaptation cannot be explained by possible perceived contrast differences caused by our high and low SF adapters. Further analysis to test the effect of individual adapters on the perceived contrast can be found in appendix Perceived contrast change after SF adaptation.

## Discussion

To our knowledge, no studies have directly explored the relationship between SF tuning and pRF sizes in humans. In this study, we provide novel empirical evidence of this relationship by incorporating an adaptation paradigm into the standard pRF mapping procedure. Recognizing that neural responsiveness to a stimulus feature diminishes after adaptation and that the most substantial change occurs in neurons typically tuned to the adapted feature (*Vautin and Berkley, 1977*), we aimed to reduce the

sensitivity and response of the high- or low-SF-tuned neuronal populations through adaptation and subsequently estimate pRF sizes.

Our results demonstrated that adapting to a band-pass SF noise pattern results in (1) a shift in the pRF size distributions in the early visual areas, and (2) a repulsive perceptual SF aftereffect on the subsequently presented pattern. The behavioral effect we found is consistent with previous adaptation studies (*Blakemore and Sutton, 1969*; *Blakemore et al., 1970*). Most importantly, our fMRI experiment showed that in multiple visual regions, ranging from V1 to V3A and V4, the spatial tuning of neuronal populations was sharpened following a low-SF adaptation and broadened after a high-SF adaptation. This provides evidence that population receptive field sizes in the early visual cortical regions are linked to the SF selectivity of visual neurons.

The relationship we showed between pRF sizes and SF tuning of neuronal populations aligns with early studies that reported a correlation between the preferred SF and classical receptive field size in the animal striate cortex (e.g. *De Valois et al., 1982*; *Movshon et al., 1978*). In addition, *Keliris et al., 2019*'s fMRI findings on humans suggested larger/smaller average single-unit receptive field (suRF) sizes for the checkerboard stimuli containing a lower/higher SF pattern, respectively. Importantly (as they also argued) their results are directly comparable to electrophysiology studies because they estimated SF-dependent BOLD response and eliminated spatial scatter of the receptive fields. This is, therefore, distinguished from the regular pRF size measurements, as clearly reflected in their results as well. Although our pRF results are not directly comparable to the suRF sizes or classical neuronal RF sizes, our findings are overall in agreement with such findings.

Nevertheless, and contrary to our findings, a previous fMRI study has found no evidence of SF-selectivity-dependent pRF size change (*Welbourne et al., 2018*). The researchers used stimuli having different chromaticities to estimate pRF sizes, with the hypothesis that the pRF sizes would change as a function of stimulus chromaticity, based on the accumulated research showing different SF tuning profiles for different color-coding pathways (e.g. *Mullen, 1985*). They found no difference between pRF estimates made with an achromatic luminance-defined grating and with either of two isoluminant chromatic gratings, and suggested that the receptive field size and SF tuning coupling does not hold at the cortical level. In light of the present study, their findings might suggest that the different SF tuning characteristics of wavelength-sensitive neurons of sub-cortical structures might not be preserved in the cortex. Alternatively, the SF-tuning and the pRF size relationship might be present in the achromatic but not the chromatic pathways. However, it is also possible that different chromaticity bars might not differentiate neuronal response, due to a likely effect of spatial attention to the bars. Previous studies established that pRFs could be estimated with occluded as well as illusory contour stimuli (*de Haas and Schwarzkopf, 2018*) and with motion-defined bar stimuli (*Hughes et al., 2019*). These findings suggest that the responses obtained from the mapping stimuli are dominated by a more complex processing rather than the specific stimulus properties. In the present study, we used the same mapping stimuli between conditions. The difference in the pRF estimations would inherently originate from SF adaptation, which not only showed a strong perceptual effect (psychophysical experiment) but also likely had a robust effect on the responsiveness of the relevant neuronal populations.

The top-up adaptation duration we used in each trial was short in comparison to some of the previous adaptation studies (*Dragoi et al., 2000*; *Liu et al., 2007*; but also see *Pavan et al., 2012*; *Tsouli et al., 2021* for other examples of sub-second adaptation to visual stimuli). Although this brief adaptation was enough to yield a strong perceptual effect behaviorally, it might raise questions about whether the observed pRF size change can possibly be related to such a short adaptation. In this regard, an alternative consideration could be that the cortical activation for the adapter and the mapping stimuli mesh together and result in distinct pRF measurements due to the possible disruptive effect of the top-up stimuli, instead of serving as adapters. Such an explanation, which rules out the adaptation effect, would not predict a decreased sensitivity of a certain group of SF selective neurons. Correspondingly, the pRF size estimations would be shifted towards the receptive field properties of the neurons selectively responding to the SF of the adapters. This scenario would predict larger pRFs in the low-SF condition and smaller pRFs in the high-SF condition, opposite to our findings. Therefore, our findings cannot be explained by such an interaction between top-up adapters and the mapping stimuli.

In addition, although the top-up adaptations in each trial were brief, it can be postulated that the effects of adaptation presented at each volume accumulate throughout the mapping sequence in our pRF experiment. This accumulation occurs because the entire visual field we measured was consistently stimulated by the adaptation stimuli, while the mapping stimulus only covered a small portion in any given trial, and it varied in size and location throughout the run. This design ensured that the effective adaptation duration consistently exceeded that of a single top-up presentation. Consequently, it can also be inferred that the adaptation effect in our fMRI experiment was likely more pronounced than what we quantified behaviorally. This inference is supported by the fact that, in the psychophysical experiment, the test stimulus consistently overlaid the same region as the adapter in every trial. Conversely, in the fMRI experiment, the confined area of the mapping bars allowed for an uninterrupted buildup of the adaptation effect across most of the visual field.

We also considered other possible confounding effects of the two SF adapter stimuli on the pRF measurements. The first of which is about the SNR. Previous work has shown that using a mapping stimulus that derives response from only a subset of neurons in a voxel results in lower SNRs and smaller pRFs (*Yildirim et al., 2018*). Both SF adapters in the present study presumably decreased the responsiveness of certain neurons within a voxel, which leads the responses to be based on a subset of neurons. If this generates disproportionate SNR changes between the two conditions, specifically lower SNR in the low-SF adapted condition, pRF size decrease in the low-SF adapted condition can possibly be attributed to the SNR differences. However, our supplementary analyses showed that this is not a concern for our pRF results. Both percent signal change time series and noise ceiling data from each vertex indicate similar SNR between the conditions.

The second considered point was about perceived contrast. Our control experiment demonstrated that the observed differences between the two adaptation conditions cannot be attributed to differences in perceived contrast. Previous research has indicated that SF adaptation increases the detection threshold of same-SF test stimuli viewed subsequently, with this effect gradually fading for test SFs that are increasingly different from the adapted SF, until the difference reaches 2 octaves (*Snowden and Hammett, 1996*). The SF we employed in our mapping stimuli falls within this reported range for both adapters, so it is reasonable to expect the perceived contrast for the mapping stimuli to be reduced under both adaptation conditions. This expectation is consistent with our supplementary analyses (See appendix Perceived contrast change after SF adaptation). Nevertheless, this perceptual deviation from the actual contrast crucially cannot account for any differences between the two adaptation conditions when the perceived contrast change is identical under both adapter conditions. Our data suggests there was no difference in perceived contrast between the high- and low-SF-adapted conditions, likely because the SF of the two adapters was equivalently distant in log space from the SF of the mapping stimuli.

While the perceived contrast following the two adapters showed no differences, a noteworthy difference was observed in the SF aftereffect. The magnitude of the perceptual aftereffect was higher after the high SF adapter, as compared to that after the low SF adapter. Similar asymmetries in bidirectional aftereffects were also reported by previous studies on SF adaptation (*Blakemore et al., 1970*), as well as other visual adaptations (*Aulet and Lourenco, 2023*; *Altan and Boyaci, 2020*). *Blakemore et al., 1970*, for example, reported that the perceptual SF shifts after low-SF adapters (between 1.5 and 3 cpd in their study) were not symmetric around the adapter SFs and were overall weaker than those after mid- and high-SF adapters. Although the reason is unclear, we speculate that it might be related to the relative locations of the adapter SFs on the contrast sensitivity function. While the adapter stimuli we used had SFs that were equally distant from the test SF in log space, they would not have equal contrast sensitivities, and the SF of the test stimuli we used did not correspond to the peak of contrast sensitivity function. This might, in turn, result in different levels of the perceptual aftereffect. Alternatively, it could also be related to possible variations in bandwidth and overlap of the SF channels.

It is plausible to expect a similar pattern between perceptual effect and pRF size change, so an equivalent asymmetry might also be present for pRF size changes caused by different adapters. However, we are unable to test this with the present data, because our experimental design only included high- and low-SF adaptation conditions (no neutral condition) to maximize scan time for each condition. Future studies could isolate the effect of each adapter stimulus by adding a neutral condition such as without adaptation or with an adapter stimulus having the same SF as the mapping

stimulus. Such a design would probably require multiple scanning sessions per participant, or maximizing the SNR by using a higher magnetic field strength. Our data also do not allow for reliable detection of the relationship between behavioral results and the pRF size change. There are two reasons for this. First, we deliberately excluded participants who did not show a strong perceptual effect, and second, we had a relatively small sample size to test such a link.

Overall, our findings, specifically spatial tuning alteration after SF adaptation, may provide insight for various domains of visual processing, including texture segmentation, cortical limits of acuity, size perception, and numerosity perception. For instance, previous research has demonstrated that adaptation to a specific SF can influence the performance accuracy in a texture segmentation task — a task that is highly interrelated with spatial resolution (*Carrasco et al., 2006*). The performance of detecting a sub-texture consisting of oriented lines that are different from its surroundings has been shown to be highest in mid-peripheral locations, while relatively low in central and more peripheral locations (*Kehrer, 1989*). *Carrasco et al., 2006* suggested that this performance variation depending on the stimulus location is due to the systematic decrease in spatial resolution (or increase in average receptive field sizes) as a function of eccentricity. They hypothesized that the detection of the texture stimulus they used would not benefit from neurons having too small (central) or too large (peripheral) receptive fields due to the scale of the texture, and thus SF adaptation would influence the performance. They found that high-SF adaptation eliminated the central performance drop. The rationale was that the high-SF adaptation would decrease the central spatial resolution, which would make it closer to the optimal spatial tuning for the task and consequently increase the task performance in central locations. Our finding provides empirical support for their explanation and aligns well with their results, showing the dynamic nature of spatial tuning in the visual field and indicating the relationship between SF tuning and spatial resolution.

Similarly, attenuating the responsiveness of neuronal populations that are irrelevant to the task can also benefit acuity. Previous work has suggested that adapting to visual stimuli with high temporal frequency (fast flicker) decreases the contribution from the magnocellular pathway that has high temporal and low spatial tuning and consequently results in an improvement in acuity due to the change in overall spatial sensitivity (*Arnold et al., 2016*). Also, it has been shown that adaptation to receding and looming motion results in an improvement in acuity (*Tagoh et al., 2022*; *Lages et al., 2017*). These findings are akin to relief from short-sightedness after some time without wearing spectacles, which can be explained by blur adaptation (*Mon-Williams et al., 1998*; *Elliott et al., 2011*), and they all convergingly suggest that decreased sensitivity of certain neuronal populations via adaptation can improve access to high-SF information which would result in an enhanced acuity. Relatedly, a decrease in the receptive field sizes of neuronal populations should also accompany this change.

Furthermore, recent psychophysical (*Bonn and Odic, 2024*), imaging (*Paul et al., 2022*), and computational (*Dakin et al., 2011*) studies presented the key role of SF in numerosity encoding. Notably, *Bonn and Odic, 2024* have demonstrated that adaptation to SFs led to various degrees of numerosity underestimation, with the largest effect observed following low-SF adaptation. This finding suggests a potential reliance of numerosity perception on neuronal populations characterized by larger receptive fields, which are more prevalent in peripheral vision. Consistent with this, their experiments also revealed that the cross-adaptation effect was evident in peripheral vision, but not in central vision.

Moreover, our results can provide additional insight into the relevance of SF channels for visual size perception. It has been proposed that some size illusions can be attributed to the SF-tuned channels in the visual system, with a particular emphasis on the role of low-SF tuned units (*Carrasco et al., 1986*; *Chen et al., 2018*). *Carrasco et al., 1986* reported that the strength of the Muller-Lyer illusion was diminished significantly after adapting to a low-SF grating and accordingly suggested that the illusory size perception occurs due to the low-SF components of the Muller-Lyer stimuli. However, this explanation has been challenged by several other studies (*Carlson et al., 1984*; *Larsen and Goldstein, 1994*). *Carlson et al., 1984*, for instance, modified the Muller-Lyer illusion so that the lines were replaced by a sequence of dots. The dots were either white dots that predominantly contained low spatial frequencies or luminance-balanced dots that contained predominantly high spatial frequencies. Even though the two types of stimuli considerably differed from each other, particularly at low spatial frequencies, they found no difference in the illusion magnitude between them. It is possible that the low-SF adaptation of *Carrasco et al., 1986* resulted in a decreased sensitivity of a neuronal group

that has relatively large receptive fields and that apparently played a key role in perceiving the illusory stimulus possibly due to the scale of the stimulus. Decreased level of activation of certain neuronal populations should be the main reason for the diminished perceptual effect, instead of the certain SF components of the stimuli. This view would not predict any difference in the illusion magnitudes of the SF-modified Muller-Lyer stimulus as in *Carlson et al., 1984* study, because the overall stimuli in both conditions would still activate the relevant large receptive field neurons with intact sensitivity.

In conclusion, we investigated the effect of SF adaptation on the spatial tuning of neuronal populations in the early visual regions. We showed that pRF sizes after high-SF adaptation were overall larger (i.e. more coarsely tuned) than those observed after low-SF adaptation. The effect was consistent between participants and also between the visual regions tested. This is evidence of a direct relationship between SF tuning and spatial resolution of the neuronal populations in humans, which could modulate downstream visual processing.

## Additional information

### Funding

| Funder | Grant reference number | Author |
| --- | --- | --- |
| Marsden Fund | 3716355 | Steven C Dakin |

The funders had no role in study design, data collection and interpretation, or the decision to submit the work for publication.

### Author contributions

Ecem Altan, Conceptualization, Data curation, Software, Formal analysis, Validation, Investigation, Visualization, Methodology, Writing – original draft, Project administration, Writing – review and editing; Catherine A Morgan, Conceptualization, Investigation, Project administration, Writing – review and editing; Steven C Dakin, Resources, Funding acquisition, Writing – review and editing, Methodology; D Samuel Schwarzkopf, Conceptualization, Supervision, Investigation, Methodology, Writing – review and editing

### Author ORCIDs

Ecem Altan ⓘ https://orcid.org/0000-0003-2902-6880
Catherine A Morgan ⓘ https://orcid.org/0000-0002-5837-8861
Steven C Dakin ⓘ https://orcid.org/0000-0002-3548-9104
D Samuel Schwarzkopf ⓘ https://orcid.org/0000-0003-3686-1622

### Ethics

The protocols and procedures were approved by the University of Auckland Human Participants Ethics Committee (reference 024231). All participants gave their informed consent prior to the experiment.

Reviewer #2 (Public review): https://doi.org/10.7554/eLife.100734.3.sa1
Reviewer #3 (Public review): https://doi.org/10.7554/eLife.100734.3.sa2
Author response https://doi.org/10.7554/eLife.100734.3.sa3

## Additional files

### Supplementary files

MDAR checklist

### Data availability

All data and code necessary to directly reproduce our reported findings (stimulus and analysis code, processed data files, statistical JASP analysis files, and more) are publicly available at https://doi.org/10.17605/OSF.IO/9KFGX (*Altan et al., 2025*). Legal and ethical regulations in Aotearoa New Zealand prohibit us from publicly sharing raw brain data of our participants. No anatomical or volumetric functional images are therefore included in this archive. We are committed to ensuring the Data

Sovereignty rights under Te Tiriti o Waitangi, prescribing the relationship between tangata whenua (indigenous people of NZ) and the Crown. Irrespective of ethnicity, no data from this study must be used beyond the purposes to which participants originally consented. The authors may share raw data on a case-by-case basis, conditional on a usage agreement and after obtaining separate ethical approval for such data use. Interested researchers may contact the senior author, Sam Schwarzkopf ( s.schwarzkopf@auckland.ac.nz).

The following dataset was generated:

| Author(s) | Year | Dataset title | Dataset URL | Database and Identifier |
|---|---|---|---|---|
| Altan E, Morgan C, Dakin S, Schwarzkopf DS | 2025 | Spatial frequency adaptation modulates population receptive field sizes | https://doi.org/10.17605/OSF.IO/9KFGX | Open Science Framework, 10.17605/OSF.IO/9KFGX |

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

## Appendix 1

### Supplementary analyses and materials

A.1 Direct comparison of pRF sizes

*Appendix 1—figure 1* presents a comparison of the adaptation conditions, illustrating the raw pRF sizes (sigma) across five cortical regions of interest (ROI) for all participants.

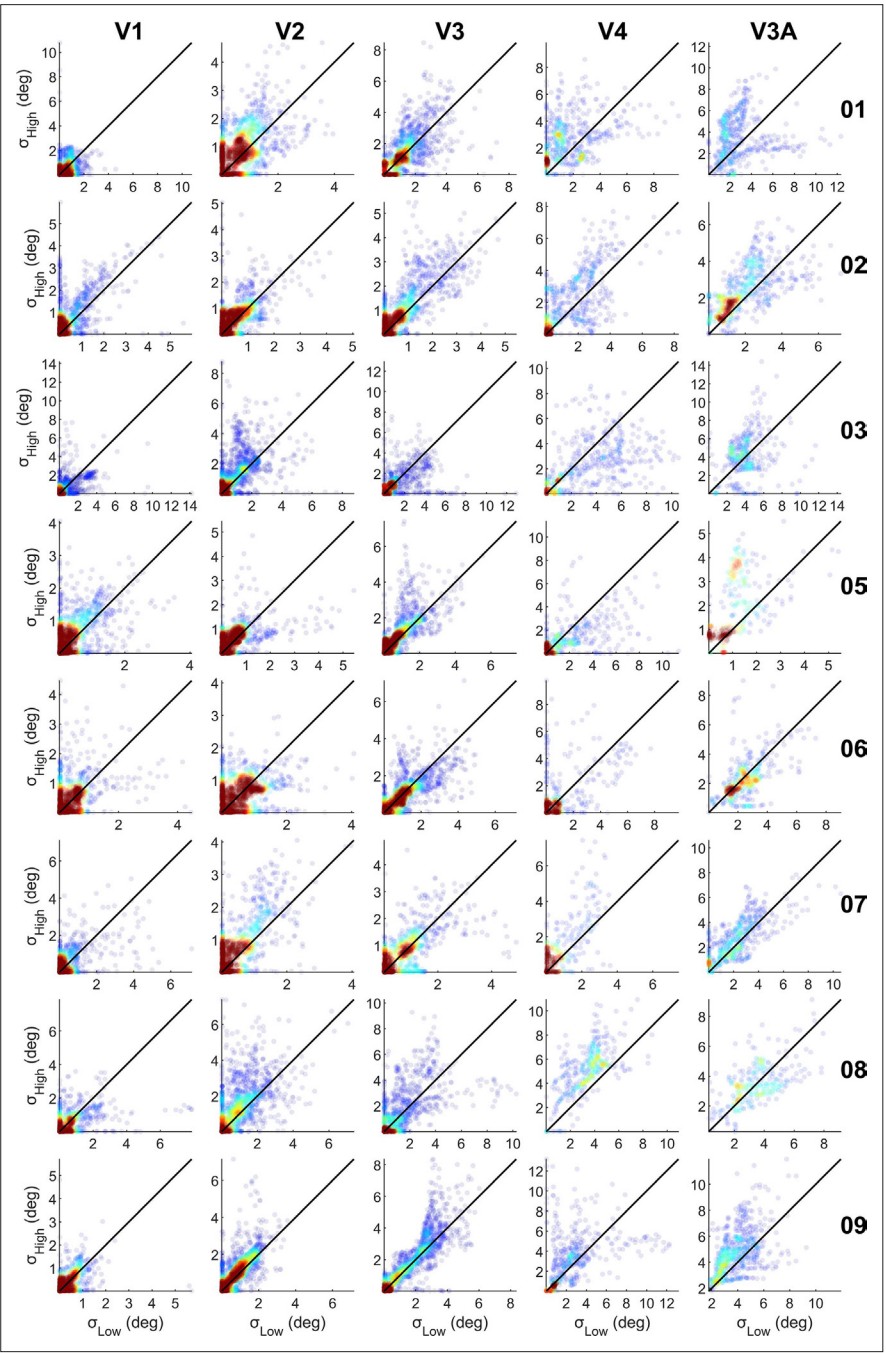

**Appendix 1—figure 1.** pRF size comparison between the two adaptation conditions for all ROIs (columns) and each participant (rows). Dots represent the pRF size, $\sigma$, of each vertex as observed in the high SF adapter condition (*y* axes) and the low SF adapter condition (*x* axes). Dots were colored based on 2D kernel density estimates. Note that the data represented here is only from the voxels that survived the denoising and $nR^2$ thresholding in both conditions. Note different scales in plots.

## A.2 Proportional pRF size change map

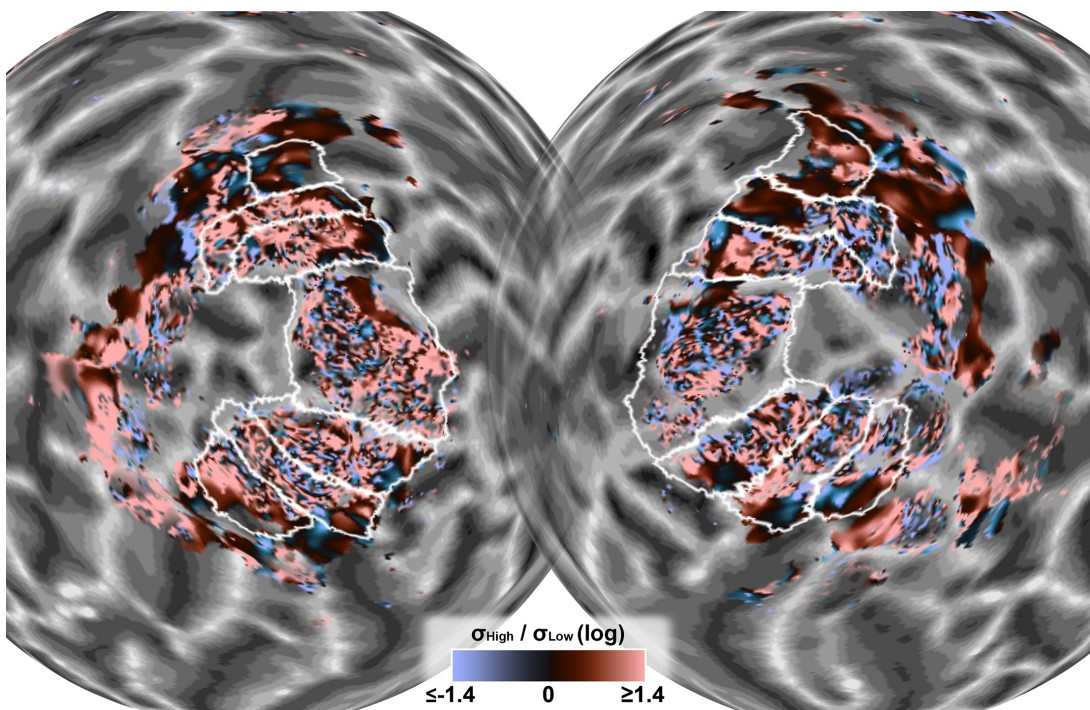

$$\sigma_{High} \,/\, \sigma_{Low} \,(\log)$$

≤-1.4 0 ≥1.4

**Appendix 1—figure 2.** Log-transformed pRF size ratio between high and low SF adapted conditions (log of H/L) for left and right hemispheres of Participant 002 (same participant as *Figure 4*). Hot colors indicate positive numbers, which represent larger pRF sizes in high SF adapted condition as compared to the low SF adapted condition; cold colors indicate the opposite. The brightness of the colors represents the magnitude of the ratios. The brightest pink (or blue) indicates pRF sizes that were at least four times as large in the high (or low) SF condition ($log_e 4 \approx 1.4$). White lines show the borders of the ROIs. Maps only show the vertices where the average of two conditions' normalized goodness of fit was above 0.2, $(nR_H^2 + nR_L^2)/2 > 0.2$. Transparency changes with respect to the proportion of the average $nR^2$. Vertices shown with opaque colors have the highest average $nR^2$ values. H: High; L: Low.

In connection with the neuronal characteristics of the visual hierarchy, the difference in pRF sizes varies considerably between ROIs and also along the eccentricity axis. Therefore, instead of the absolute pRF size difference (as in *Figure 5b*), we visualized the proportional difference between the two conditions.

To visualize the proportional pRF size difference between the two conditions, we divided the pRF sizes (sigmas) of the high SF adapted condition by the pRF sizes of the low SF adapted condition ($\sigma_{High}/\sigma_{Low}$). The sigma ratios were then logarithmically transformed and projected onto the cortical surface map.

*Appendix 1—figure 2* shows the ratio map generated for Participant 02. For this figure, we thresholded the vertices based on the average normalized goodness of fit, and we removed those smaller than or equal to 0.2, $(nR_{High}^2 + nR_{Low}^2)/2 \leq 0.2$. The figure illustrates that the proportional change in the pRF sizes between the two adaptation conditions is mostly similar between the regions.

A.3 Signal-to-noise ratio

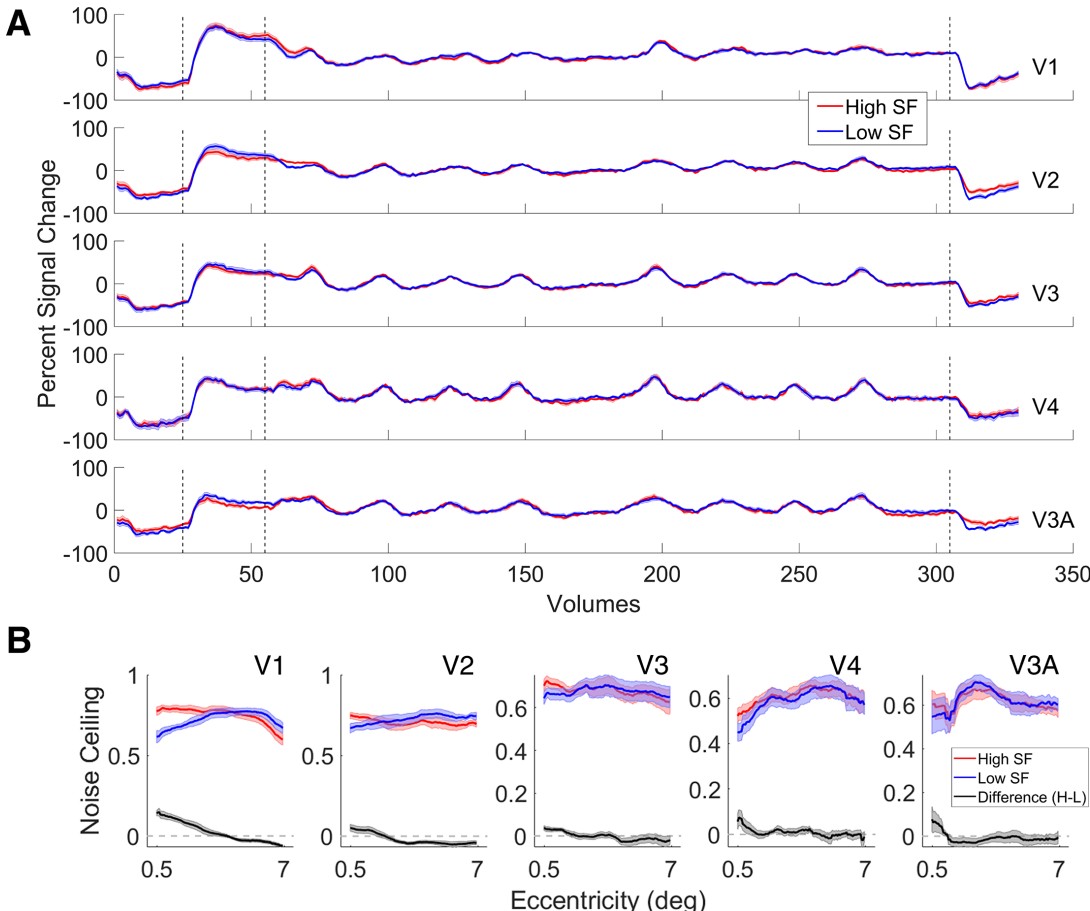

**Appendix 1—figure 3.** Signal-to-noise ratio comparison between two SF adapter conditions. (**A**) Time series of percent signal change for two adaptation conditions in each ROI, averaged across eight participants. Regions were annotated on the right side of each panel. Red and blue lines respectively represent high and low SF adapted conditions. The shaded regions indicate ±1 standard error between the participants. The vertical dashed lines, respectively from left to right, indicate the end of the initial baseline period (volume 25), the end of the initial adaptation period (volume 55), and the start of the final baseline period (volume 305). The volumes between the second and third dashed lines correspond to the mapping sequence. (**B**) Median of eccentricity binned noise ceiling values, averaged across participants (N=8). The average of the two conditions was used for eccentricity binning. The shaded area represents the standard error of the mean between participants.

One may argue that the signal-to-noise ratio (SNR) could vary between the two adapters, potentially affecting the pRF estimations. To examine this, we repeated the preprocessing steps subsequent to the surface projection (refer to 'Preprocessing'), with two alterations: (1) the time series were detrended but not $z$-normalized, and (2) none of the volumes were discarded prior to merging the hemispheres of the same condition.

We then removed the vertices having a normalized $R^2$ smaller than 0.2 as we also did in the pRF analysis. Remaining vertices were averaged within each ROI, and the time courses for both conditions were plotted for all participants. *Appendix 1—figure 3a* shows the time courses in each ROI, averaged across participants. A visual inspection of the figure suggests that the signal amplitudes from the two adaptation conditions highly match, especially during the pRF mapping sequence.

The figure shows small differences between the two conditions during the adaptation periods. We believe this is because of the different spatial frequency content of the adapter stimuli *Singh et al., 2000*. Besides, slight differences during the initial and the final baseline periods are likely to be due to the overall signal modulations with regard to the variations in signal to the visual

stimulation relative to that to the baseline. However, these small differences before and after the mapping sequence are not likely to have an effect on our results.

Since the noise ceiling, the maximum achievable goodness-of-fit, is also indicative of the signal-to-noise ratio, we also compared noise ceiling values in two conditions along the eccentricity axis. To do so, we binned the noise ceiling data into 100 1° wide eccentricity bins using a sliding window approach. We used the average of the two conditions for eccentricity selection. Median values of each bin were calculated for both conditions and averaged across participants. *Appendix 1— figure 3b* represents the average noise ceiling data from the two SF adaptation conditions (red: high SF; blue: low SF) as well as the difference between the noise ceiling values (black), binned into eccentricity bands. Overall, the noise ceiling plots across eccentricity are similar between the two conditions. The black line, the difference between the conditions, is consistently close to zero with an exception in central V1. The noise ceiling in central V1, which mostly consists of smaller pRFs, is larger in the high SF adaptation condition than in the low SF adaptation condition. Although the reason behind this difference is unclear, it does not follow the pRF size change we observed in this region and, therefore, cannot account for our main findings.

## A.4 Vertex thresholding

After filtering out the artifactual vertices and the vertices with a low goodness-of-fit, we did not apply further removal of vertices from our analyses, as explained in the main text. However, there might be a possible systematic bias where the observed responses from the two adapter conditions were derived from different sets of vertices and, thereby, different regions of the cortex. To control for this possible bias, we present the pRF size difference plots using the same vertices between the two conditions in *Appendix 1—figure 4a*. This figure was obtained in the same way as *Figure 5b*, with the only exception that the thresholded vertices in one condition were removed in another. The figure shows almost identical results as in *Figure 5b*, indicating that the observed pRF size difference is not a result of such a bias.

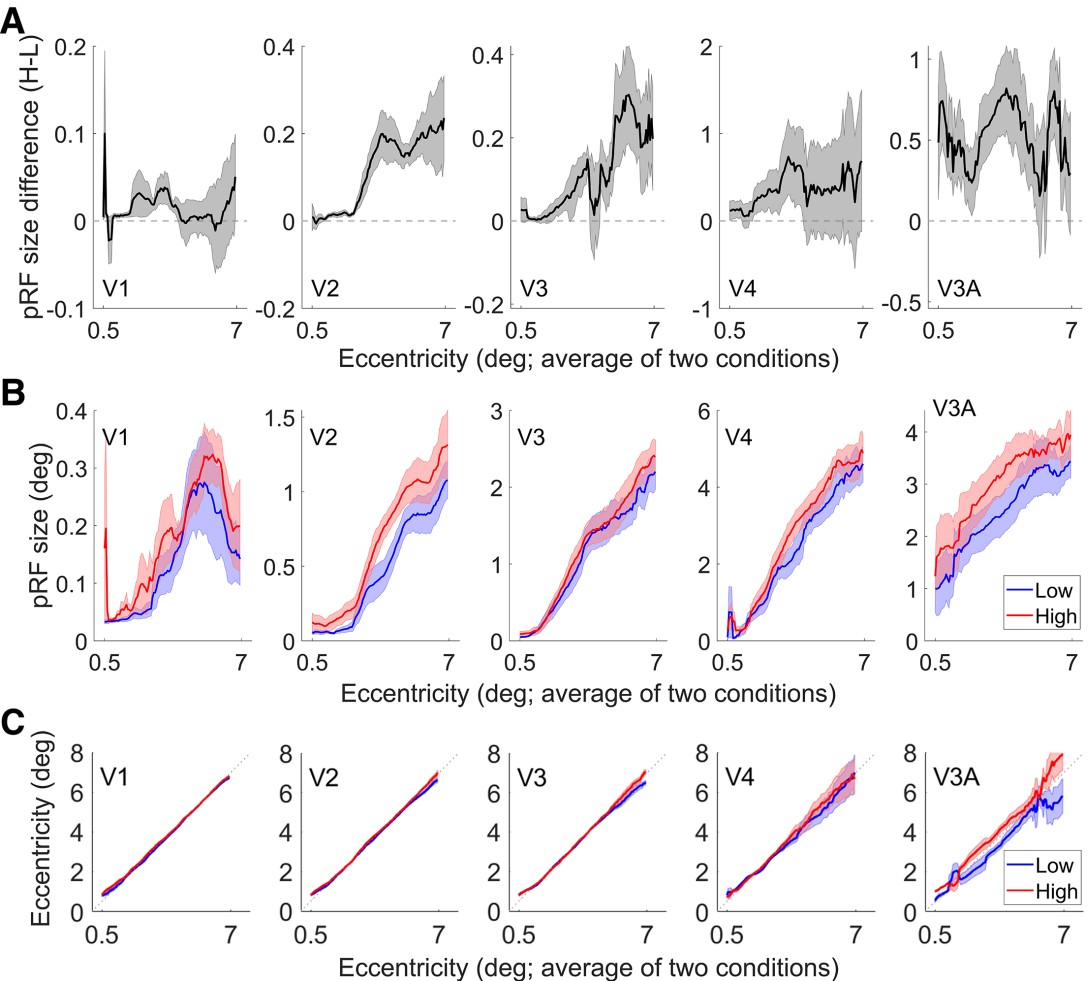

**Appendix 1—figure 4.** pRF size difference using the same vertices, pRF sizes and eccentricities as a function of eccentricity. (**A**) Median of eccentricity binned pRF size differences, averaged across participants (N=8), using the same set of vertices in two conditions. (**B**) Median of eccentricity binned sigma values for both high and low SF adaptation conditions, averaged across participants (N=8; Note different scales between visual regions in panels **A** and **B**). (**C**) Median of eccentricity binned eccentricity values, averaged across participants (N=8). For all panels, the average of the two adaptation conditions was used for eccentricity selection (*x*-axes). The shaded regions represent the standard error of the mean between participants.

## A.5 Does SF adaptation alter other pRF estimates?

To test if our SF adapters resulted in any systematic shift in pRF eccentricities, we binned the eccentricity values of vertices from two conditions into 100 eccentricity bins, using a sliding window approach. We used the average of the two conditions for eccentricity selection. The median values of each bin were calculated for both conditions and averaged across participants. *Appendix 1—figure 4c* demonstrates the comparison between the two adaptation conditions. The figure clearly shows that pRF eccentricities in two adaptation conditions do not differ from each other, especially in regions between V1 and V4. There is a small difference in V3A, but this should be interpreted with caution due to the noisy data in V3A, related to the small number of vertices and large pRFs.

We also observed no change in the polar angle estimates. We present raw comparisons of polar angle, eccentricity, and sigma estimates for each participant in the shared data repository.

## A.6 Does the perceptual effect differ between the two visual hemifields?

We performed an additional statistical analysis to identify whether there is an asymmetry in the adaptation effect between the two visual hemifields. To perform this analysis, we repeated the fitting procedure, but this time we pooled the data separately for two visual hemifields. We had six

psychometric functions and therefore six PSEs for each participant (two visual hemifields × three adaptation conditions). We performed a repeated measures ANOVA with two independent variables and bias-corrected, log-transformed SF ratio (perceived/actual SF) values as our dependent variable. Results suggested that there was a significant main effect of adaptation SF ($F(1, 9) = 40$, $p < 0.001$), and there was no significant main effect of visual hemifield ($F(1, 9) = 0.9$, $p = 0.4$). There was also no significant interaction ($F(1, 9) = 0.03$, $p = 0.9$). Results can be seen in *Appendix 1—figure 5*.

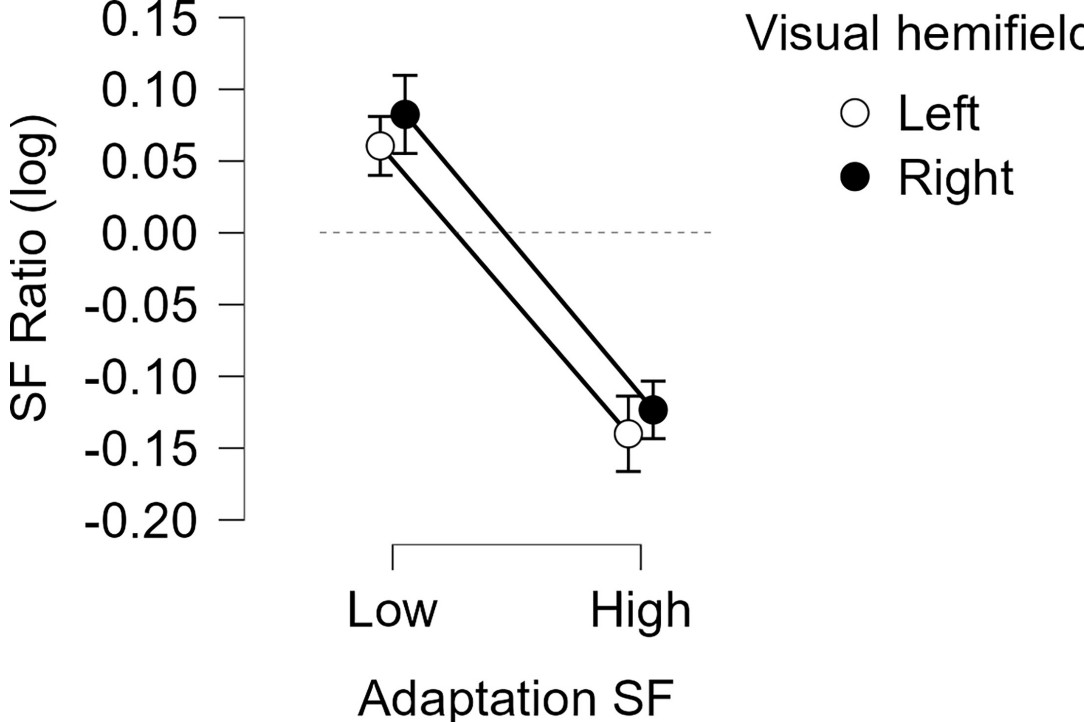

**Appendix 1—figure 5.** Log-transformed and bias-corrected SF ratios (perceived/actual SF) were plotted against the adaptation conditions, separately for left and right visual fields. Positive (negative) values of $y$-axis indicate an increase (decrease) in perceived SF and zero means no perceptual deviation from the actual SF. Error bars represent standard error of the mean across participants (N=10).

## A.7 Eye tracking data

To test whether the pupil size or the distance between the eye and the fixation point (i.e. screen center) differs between the two adaptation conditions, we first decoupled the eye positions according to the SF adapter condition they were shifted towards. We then pooled the pupil size and eye position data of each participant from all runs. We compared the median values of the pooled data between the two adaptation conditions (See *Appendix 1—figure 6*). We found that the paired samples $t$-tests for the distance ($t(7) = -0.14$, $p = 0.9$) and the pupil size ($t(7) = 0.1$, $p = 0.9$) did not show significant differences depending on the SF adapter conditions. We also performed Bayesian hypothesis tests (paired samples $t$- tests) for the distance and pupil size. We used the default uninformed prior distribution and parameters in JASP. Both tests showed anecdotal support for the null hypothesis which suggests no difference between the adaptation conditions ($BF_{null} = 2.95$ for the fixation distance and $BF_{null} = 2.96$ for the pupil size; error = 0.003 for both)

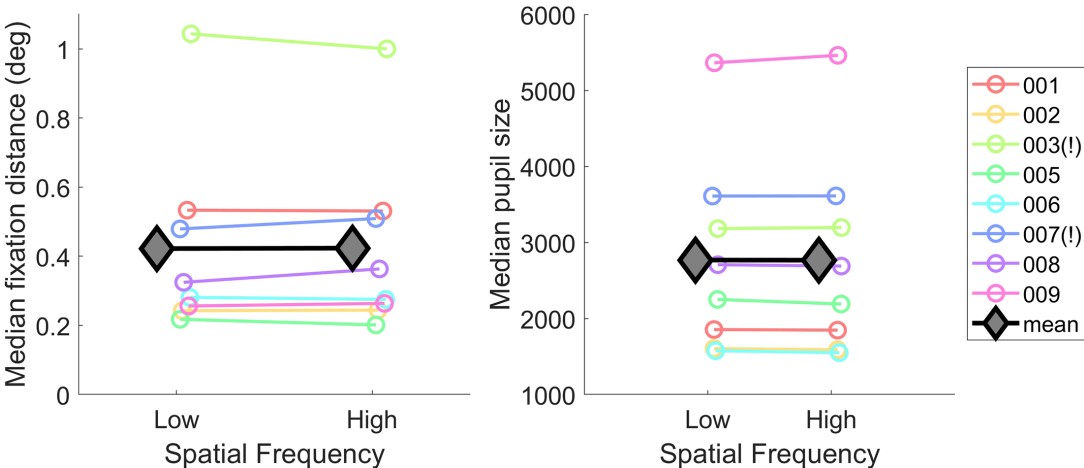

**Appendix 1—figure 6.** Distance between the eye position and the fixation point (left) and pupil size (right) plotted against the two adaptation conditions. Colored circles represent each participant and gray diamonds show their mean. The exclamation marks on the legend items denote the participants who were excluded from the statistical analyses.

## A.8 Perceived contrast change after SF adaptation

Critically, we presented that the SF adapters do not differ from each other in terms of their effect on perceived contrast in Can perceived contrast explain the findings? Since our data can provide further information on whether SF adapters influence the perceived contrast of the subsequent stimuli, we performed additional statistical analyses. We used the same log-transformed and bias-corrected PSE values that were described in 'Results' and applied two one-sample $t$-tests for each adapter condition. Results showed that both low- and high-SF adapters resulted in a significant decrease in the perceived contrast of the following mid-SF test stimulus ($t(9) = -4.5$, $p_{bonf} < 0.01$ for low SF; $t(8) = -4$, $p_{bonf} < 0.01$ for high SF). Bayesian one-sample $t$-test hypothesis testings with uninformed prior with the default parameters also showed strong evidence for the perceived contrast change after both low ($BF_{10} = 28.9$) and high ($BF_{10} = 14.1$) SF adapters. Both tests showed very robust effects when tested with various prior distribution (Cauchy) widths. Results indicate a clear decrease in perceived contrast as a result of SF adaptation, which is consistent with the literature *Snowden and Hammett, 1996*.

However, the magnitude of these effects should be interpreted with caution because our experiment was not designed to answer this question and therefore is suboptimal for testing the magnitude of perceived contrast shift after adaptation. Given that the reference stimulus had a maximum contrast level of 100%, and the test stimulus could not exceed this limit, any random key presses when both stimuli had equivalent contrast would lead to a small but artificial decrease in the measured perceived contrast. This decrease would be roughly half of the smallest step size, which is 0.05 as clearly observable in the no-adaptation condition in *Figure 6b*. Even though this artificial decrease should not necessarily influence the individual SF adaptation conditions —given that the perceived contrasts under these conditions are considerably distant from the point where the reference and test stimuli are equal— the applied bias correction would introduce a minor impact.

