## [Editor Report · eLife Assessment]

This well-designed study combining psychophysical and fMRI data presents a **valuable** finding regarding how adaptation alters spatial frequency processing in the cortex. The evidence supporting the claims of the authors is **solid**, although inclusion of more participants and better quality of the fMRI data would have strengthened the study. The study will be of interest to cognitive and perceptual neuroscientists working on human and non-human primates.

---

## [Referee Report · Reviewer #2 (Public review)]

The revised manuscript by Altan et al. includes some real improvements to the visualizations and explanations of the authors' thesis statement with respect to fMRI measurements of pRF sizes. In particular, the deposition of the paper's data has allowed me to probe and refine several of my previous concerns. While I still have major concerns about how the data are presented in the current draft of the manuscript, my skepticism about data quality overall has been much alleviated. Note that this review focuses almost exclusively on the fMRI data as I was satisfied with the quality of the psychophysical data and analyses in my previous review.

Major Concerns

(I) Statistical Analysis

In my previous review, I raised the concern that the small sample size combined with the noisiness of the fMRI data, a lack of clarity about some of the statistics, and a lack of code/data likely combine to make this paper difficult or impossible to reproduce as it stands. The authors have since addressed several aspects of this concern, most importantly by depositing their data. However their response leaves some major questions, which I detail below.

First of all, the authors claim in their response to the previous review that the small sample size is not an issue because large samples are not necessary to obtain "conclusive" results. They are, of course, technically correct that a small sample size can yield significant results, but the response misses the point entirely. In fact, small samples are more likely than large samples to erroneously yield a significant result (Button et al., 2013, DOI:10.1038/nrn3475), especially when noise is high. The response by the authors cites Schwarzkopf & Huang (2024) to support their methods on this front. After reading the paper, I fail to see how it is at all relevant to the manuscript at hand or the criticism raised in the previous review. Schwarzkopf & Huang propose a statistical framework that is narrowly tailored to situations where one is already certain that some phenomenon (like the adaptation of pRF size to spatial frequency) either always occurs or never occurs. Such a framework is invalid if one cannot be certain that, for example, pRF size adapts in 98% of people but not the remaining 2%. Even if the paper were relevant to the current study, the authors don't cite this paper, use its framework, or admit the assumptions it requires in the current manuscript. The observation that a small dataset can theoretically lead to significance under a set of assumptions not appropriate for the current manuscript is not a serious response to the concern that this manuscript may not be reproducible.

To overcome this concern, the authors should provide clear descriptions of their statistical analyses and explanations of why these analyses are appropriate for the data. Ideally, source code should be published that demonstrates how the statistical tests were run on the published data. (I was unable to find any such source code in the OSF repository.) If the effects in the paper were much stronger, this level of rigor might not be strictly necessary, but the data currently give the impression of being right near the boundary of significance, and the manuscript's analyses needs to reflect that. The descriptions in the text were helpful, but I was only able to approximately reproduce the authors analyses based on these descriptions alone. Specifically, I attempted to reproduce the Mood's median tests described in the second paragraph of section 3.2 after filtering the data based on the criteria described in the final paragraph of section 3.1. I found that 7/8 (V1), 7/8 (V2), 5/8 (V3), 5/8 (V4), and 4/8 (V3A) subjects passed the median test when accounting for the (40) multiple comparisons. These results are reasonably close to those reported in the manuscript and might just differ based on the multiple comparisons strategy used (which I did not find documented in the manuscript). However, Mood's median test does not test the direction of the difference-just whether the medians are different-so I additionally required that the median sigma of the high-adapted pRFs be greater than that of the low-adapted pRFs. Surprisingly, in V1 and V3, one subject each (not the same subject) failed this part of the test, meaning that they had significant differences between conditions but in the wrong direction. This leaves 6/8 (V1), 7/8 (V2), 4/8 (V3), 5/8 (V4), and 4/8 (V3A) subjects that appear to support the authors' conclusions. As the authors mention, however, this set of analyses runs the risk of comparing different parts of cortex, so I also performed Wilcox signed-rank tests on the (paired) vertex data for which both the high-adapted and low-adapted conditions passed all the authors' stated thresholds. These results largely agreed with the median test (only 5/8 subjects significant in V1 but 6/8 in in V3A, other areas the same, though the two tests did not always agree which subjects had significant differences). These analyses were of course performed by a reviewer with a reviewer's time commitment to the project and shouldn't be considered a replacement for the authors' expertise with their own data. If the authors think that I have made a mistake in these calculations, then the best way to refute them would be to publish the source code they used to threshold the data and to perform the same tests.

Setting aside the precise values of the relevant tests, we should also consider whether 5 of 8 subjects showing a significant effect (as they report for V3, for example) should count as significant evidence of the effect? If one assumes, as a null hypothesis, that there is no difference between the two conditions in V3 and that all differences are purely noise, then a binomial test across subjects would be appropriate. Even if 6 of 8 subjects show the effect, however (and ignoring multiple comparisons), the p-value of a one-sided binomial test is not significant at the 0.05 level (7 of 8 subjects is barely significant). Of course, a more rigorous way to approach this question could be something like an ANOVA, and the authors use an ANOVA analysis of the medians in the paragraph following their use of Mood's median test. However, ANOVA assumes normality, and the authors state in the previous paragraph that they employed Mood's median test because "the distribution of the pRF sizes is zero-bounded and highly skewed" so this choice does not make sense. The Central Limits Theorem might be applied to the medians in theory, but with only 8 subjects and with an underlying distribution of pRF sizes that is non-negative, the relevant data will almost certainly not be normally distributed. These tests should probably be something like a Kruskal-Wallis ANOVA on ranks.

All of the above said, my intuition about the data is currently that there are significant changes to the adapted pRF size in V2. I am not currently convinced that the effects in other visual areas are significant, and I suspect that the paper would be improved if authors abandoned their claims that areas other than V2 show a substantial effect. Importantly, I don't think this causes the paper to lose any impact-in fact, if the authors agree with my assessments, then the paper might be improved by focusing on V2. Specifically, the authors' already discuss psychophysical work related to the perception of texture on pages 18 and 19 and link it to their results. V2 is also implicated in the perception of texture (see, for example, Freeman et al., 2013; DOI:10.1038/nn.3402; Ziemba et al., 2016, DOI:10.1073/pnas.1510847113; Ziemba et al., 2019; DOI:10.1523/JNEUROSCI.1743-19.2019) and so would naturally be the part of the visual cortex where one might predict that spatial frequency adaptation would have a strong effect on pRF size. This neatly connects the psychophysical and imaging sides of this project and could make a very nice story out of the present work.

(II) Visualizations

The manuscript's visual evidence regarding the pRF data also remains fairly weak (but I found the pRF size comparisons in the OSF repository and Figure S1 to be better evidence-more in the next paragraph). The first line of the Results section still states, "A visual inspection on the pRF size maps in Figure 4c clearly shows a difference between the two conditions, which is evident in all regions." As I mentioned in my previous review, I don't agree with this claim (specifically, that it is clear). My impression when I look at these plots is of similarity between the maps, and, where there is dissimilarity, of likely artifacts. For example, the splotch of cortex near the upper vertical meridian (ventral boundary) of V1 that shows up in yellow in the upper plot but not the lower plot also has a weirdly high eccentricity and a polar angle near the opposite vertical meridian: almost certainly not the actual tuning of that patch of cortex. If this is the clearest example subject in the dataset, then the effect looks to me to be very small and inconsistently distributed across the visual areas. That said, I'm not convinced that the problem here is the data-rather, I think it's just very hard to communicate a small difference in parameter tuning across a visual area using this kind of side-by-side figure. I think that Figure S2, though noisy (as pRF maps typically are), is more convincing than Figure 4c, personally. For what it's worth, when looking at the data myself, I found that plotting log(𝜎(H) / 𝜎(L)), which will be unstable when noise causes 𝜎(H) or 𝜎(L) to approach zero, was less useful than plotting plotting (𝜎(H) - 𝜎(L)) / (𝜎(H) + 𝜎(L)). This latter quantity will be constrained between -1 and 1 and shows something like a proportional change in the pRF size (and thus should be more comparable across eccentricity).

In my opinion, the inclusion of the pRF size comparison plots in the OSF repository and Figure S1 made a stronger case than any of the plots of the cortical surface. I would suggest putting these on log-log plots since the distribution of pRF size (like eccentricity) is approximately exponential on the cortical surface. As-is, it's clear in many plots that there is a big splotch of data in the compressed lower left corner, but it's hard to get a sense for how these should be compared to the upper right expanse of the plots. It is frequently hard to tell whether there is a greater concentration of points above or below the line of equality in the lower left corner as well, and this is fairly central to the paper's claims. My intuition is that the upper right is showing relatively little data (maybe 10%?), but these data are very emphasized by the current plots.The authors might even want to consider putting a collection of these scatter-plots (or maybe just subject 007, or possible all subjects' pRFs on a single scatter-plot) in the main paper and using these visualizations to provide intuitive supporting for the main conclusions about the fMRI data (where the manuscript currently use Figure 4c for visual intuition).

Minor Comments

(1) Although eLife does not strictly require it, I would like to see more of the authors' code deposited along with the data (especially the code for calculating the statistics that were mentioned above). I do appreciate the simulation code that the authors added in the latest submission (largely added in response to my criticism in the previous reviews), and I'll admit that it helped me understand where the authors were coming from, but it also contains a bug and thus makes a good example of why I'd like to see more of the authors' code. If we set aside the scientific question of whether the simulation is representative of an fMRI voxel (more in Minor Comment 5, below), Figures 1A and the "AdaptaionEffectSimulated.png" file from the repository (https://osf.io/d5agf) imply that only small RFs were excluded in the high-adapted condition and only large RFs were excluded in the low-adapted condition. However, the script provided (SimlatePrfAdaptation.m: https://osf.io/u4d2h) does not do this. Lines 7 and 8 of the script set the small and large cutoffs at the 30th and 70th percentiles, respectively, then exclude everything greater than the 30th percentile in the "Large RFs adapted out" condition (lines 19-21) and exclude anything less than the 70th percentile in the "Small RFs adapted out" condition (lines 27-29). So the figures imply that they are representing 70% of the data but they are in fact representing only the most extreme 30% of the data. (Moreover, I was unable to run the script because it contains hard-coded paths to code in someone's home directory.) Just to be clear, these kinds of bugs are quite common in scientific code, and this bug was almost certainly an honest mistake.

(2) I also noticed that the individual subject scatter-plots of high versus low adapted pRF sizes on the OSF seem to occasionally have a large concentration of values on the x=0 and y=0 axes. This isn't really a big deal in the plots, but the manuscript states that "we denoised the pRF data to remove artifactual vertices where at least one of the following criteria was met: (1) sigma values were equal to or less than zero ..." so I would encourage the authors to double-check that the rest of their analysis code was run with the stated filtering.

(3) The manuscript also says that the median test was performed "on the raw pRF size values". I'm not really sure what the "raw" means here. Does this refer to pRF sizes without thresholding applied?

(4) The eccentricity data are much clearer now with the additional comments from the authors and the full set of maps; my concerns about this point have been met.

(5) Regarding the simulation of RFs in a voxel (setting aside the bug), I will admit both to hoping for a more biologically-grounded situation and to nonetheless understanding where the authors are coming from based on the provided example. What I mean by biologically-grounded: something like, assume a 2.5-mm isotropic voxel aligned to the surface of V1 at 4{degree sign} of eccentricity; the voxel would span X to Y degrees of eccentricity, and we predict Z neurons with RFs in this voxel with a distribution of RF sizes at that eccentricity from [reference], etc. eventually demonstrating a plausible pRF size change commensurate to the paper's measurements. I do think that a simulation like this would make the paper more compelling, but I'll acknowledge that it probably isn't necessary and might be beyond the scope here.

---

## [Referee Report · Reviewer #3 (Public review)]

This is a well-designed study examining an important, surprisingly understudied question: how does adaptation affect spatial frequency processing in human visual cortex? Using a combination of psychophysics and neuroimaging, the authors test the hypothesis that spatial frequency tuning is shifted to higher or lower frequencies, depending on preadapted state (low or high s.f. adaptation). They do so by first validating the phenomenon psychophysically, showing that adapting to 0.5 cpd stimuli causes an increase perceived s.f., and 3.5 cpd causes a relative decrease in perceived s.f. Using the same stimuli, they then port these stimuli to a neuroimaging study, in which population receptive fields are measured under high and low spatial frequency adaptation states. They find that adaptation changes pRF size, depending on adaptation state: adapting to high s.f. led to broader overall pRF sizes across early visual cortex, whereas adapting to low s.f. led to smaller overall pRF sizes. Finally the authors carry out a control experiment to psychophysically rule out the possibility that the perceived contrast change w/ adaptation may have given rise to these imaging results (doesn't appear to be the case). All in all, I found this to be a good manuscript: the writing is taut, and the study is well designed.

---

## [Author Response]

The following is the authors’ response to the original reviews.

**Reviewer #1 (Public Review):**

We thank the reviewer for their careful evaluation and positive comments.

Adaptation paradigm“why is it necessary to use an *adaptation* paradigm to study the link between SF tuning and pRF estimation? Couldn't you just use pRF bar stimuli with varying SFs?”

We thank the reviewer for this question. First, by using adaptation we can infer the correspondence between the perceptual and the neuronal adaptation to spatial frequency. We couldn’t draw any inference about perception if we only varied the SF inside the bar. More importantly, while changing the SF inside the bar might help drive different neuronal populations, this is not guaranteed. As we touched on in our discussion, responses obtained from the mapping stimuli are dominated by complex processing rather than the stimulus properties alone. A considerable proportion of the retinotopic mapping signal is probably simply due to spatial attention to the bar (de Haas & Schwarzkopf, 2018; Hughes et al., 2019). So, adaptation is a more targeted way to manipulate different neuronal populations.

Other pRF estimates: polar angle and eccentricity

We included an additional plot showing the polar angle for both adapter conditions (Figure S4), as well as participant-wise scatter plots comparing raw pRF size, eccentricity, and polar angle between two adapter conditions (available in shared data repository). In line with previous work on the reliability of pRF estimates (van Dijk, de Haas, Moutsiana, & Schwarzkopf, 2016; Senden, Reithler, Gijsen, & Goebel, 2014), both polar angle and eccentricity maps are very stable between the two adaptation conditions.

Variability in pRF size change

As the reviewer pointed out, the pRF size changes show some variability across eccentricities, and ROIs (Figure 5A and 5B). It is likely that the variability could relate to the varying tuning properties of different regions and eccentricities for the specific SF we used in the mapping stimulus. So one reason V2 is most consistent could be that the stimulus is best matched for the tuning there. However, what factors contribute to this variability is an interesting question that will require further study.

Other recommendations

We have addressed the other recommendations of the reviewer with one exception. The reviewer suggested we should comment on the perceived contrast decrease after SF adaptation (as seen in Figure 6B) *in the main text*. However, since we refer the readers to the supplementary analyses (Supplementary section S8) where we discuss this in detail, we chose to keep this aspect unchanged to avoid overcomplicating the main text.

**Reviewer #2 (Public Review):**

We thank the reviewer for their comments - we improved how we report key findings which we hope will clarify matters raised by the reviewer.

RF positions in a voxel

The reviewer’s comments suggest that they may have misunderstood the diagram (Figure 1A) illustrating the theoretical basis of the adaptation effect, likely due to us inadvertently putting the small RFs in the middle of the illustration. We changed this figure to avoid such confusion.

Theoretical explanation of adaptation effect

The reviewer’s explanation for how adaptation should affect the size of pRF averaging across individual RFs is incorrect. When selecting RFs from a fixed range of semi-uniformly distributed positions (as in an fMRI voxel), the average position of RFs (corresponding to pRF position) is naturally near the center of this range. The average size (corresponding to pRF size) reflects the visual field coverage of these individual RFs. This aggregate visual field coverage thus also reflects the individual sizes. When large RFs have been adapted out, this means the visual field coverage at the boundaries is sparser, and the aggregate pRF is therefore smaller. The opposite happens when adapting out the contribution of small RFs. We demonstrate this with a simple simulation at this OSF link: https://osf.io/ebnky/. The pRF size of the simulated voxels illustrate the adaptation effect should manifest precisely as we hypothesized.

Figure S2

It is not actually possible to compare R^2^ between regions by looking at Figure S2 because it shows the pRF size change, not R^2^. Therefore, the arguments Reviewer #2 made based on their interpretation of the figure are not valid. Just as the reviewer expected, V1 is one of the brain regions with good pRF model fits. We included normalized and raw R^2^ maps to make this more obvious to the readers.

V1 appeared essentially empty in that plot primarily due to the sigma threshold we selected, which was unintentionally more conservative than those applied in our analyses and other figures. We apologize for this mistake. We corrected it in the revised version by including a plot with the appropriate sigma threshold.

Thresholding details

Thresholding information was included in our original manuscript; however, we included more information in the figure captions to make it more obvious.

2D plots replaced histograms

We thank the reviewer for this suggestion. The original manuscript contained histograms showing the distribution of pRF size for both adaptation conditions for each participant and visual area (Figure S1). However, we agree that 2D plots better communicate the difference in pRF parameters between conditions. So we moved the histogram plots to the online repository, and included scatter plots with a color scheme revealing the 2D kernel density.

We chose to implement 2D kernel density in scatter plots to display the distribution of individual pRF sizes transparently.

(proportional) pRF size-change map

The reviewer requests pRF size difference maps. Figure S2 in fact demonstrates the proportional difference between the pRF sizes of the two adaptation conditions. Instead of simply taking the difference, we believe showing the proportional change map is more sensible because overall pRF size varies considerably between visual regions. We explained this more clearly in our revision.

pRF eccentricity plot

“I suspect that the difference in PRF size across voxels correlates very strongly with the difference in eccentricity across voxels.”

Our original manuscript already contained a supplementary plot (Figure S4 B, now Figure S4 C) comparing the eccentricity between adapter conditions, showing no notable shift in eccentricities except in V3A - but that is a small region and the results are generally more variable. In addition, we included participant-wise plots in the online repository, presenting raw comparisons of pRF size, eccentricity, and polar angle estimates between adaptation conditions. These 2D plots provide further evidence that the SF adapters resulted in a change in pRF size, while eccentricity and polar angle estimates did not show consistent differences.

To the reviewer’s point, even if there were an appreciable shift in eccentricity between conditions (as they suggest may have happened for the example participant we showed), this does not mean that the pRF size effect is “due [...] to shifts in eccentricity.” Parameters in a complex multi-dimensional model like the pRF are not independent. There is no way of knowing whether a change in one parameter is causally linked with a change in another. We can only report the parameter estimates the model produces.

In fact, it is conceivable that adaptation causes both: changes in pRF size *and* eccentricity. If more central or peripheral RFs tend to have smaller or larger RFs, respectively, then adapting out one part of the distribution will shift the average accordingly. However, as we already established, we find no compelling evidence that pRF eccentricity changes dramatically due to adaptation, while pRF size does.

Other recommendations

We have addressed the other recommendations of the reviewer, except for the y-axis alignment. Different regions in the visual hierarchy naturally vary substantially in pRF size. Aligning axes would therefore lead to incorrect visual inferences that (1) the absolute pRF sizes between ROIs are comparable, and (2) higher regions show the effect most

prominently. However, for clarity, we now note this scale difference in our figure captions. Finally, as mentioned earlier, we also present a proportional pRF size change map to enable comparison of the adaptation effect between regions.

**Reviewer #3 (Public Review):**

We thank the reviewer for their comments.

pRF model

Top-up adapters were not modelled in our analyses because they are shared events in all TRs, critically also including the “blank” periods, providing a constant source of signal. Therefore modelling them separately cannot meaningfully change the results. However, the reviewer makes a good suggestion that it would be useful to mention this in the manuscript, so we added a discussion of this point in Section 3.1.5.

pRF size vs eccentricity

We added a plot showing pRF size in the two adaptation conditions (in addition to the pRF size difference) as a function of eccentricity.

Correlation with behavioral effect

In the original manuscript, we pointed out why the correlation between the magnitude of the behavioral effect and the pRF size change is not an appropriate test for our data. First, the reviewer is right that a larger sample size would be needed to reliably detect such a between-subject correlation. More importantly, as per our recruitment criteria for the fMRI experiment, we did not scan participants showing weak perceptual effects. This limits the variability in the perceptual effect and makes correlation inapplicable.